# Primordial aqueous alteration recorded in water-soluble organic molecules from the carbonaceous asteroid (162173) Ryugu

We report primordial aqueous alteration signatures in water-soluble organic molecules from the carbonaceous asteroid (162173) Ryugu by the Hayabusa2 spacecraft of JAXA. Newly identified low-molecular-weight hydroxy acids (HO-R-COOH) and dicarboxylic acids (HOOC-R-COOH), such as glycolic acid, lactic acid, glyceric acid, oxalic acid, and succinic acid, are predominant in samples from the two touchdown locations at Ryugu. The quantitative and qualitative profiles for the hydrophilic molecules between the two sampling locations shows similar trends within the order of ppb (parts per billion) to ppm (parts per million). A wide variety of structural isomers, including α- and β-hydroxy acids, are observed among the hydrophilic molecules. We also identify pyruvic acid and dihydroxy and tricarboxylic acids, which are biochemically important intermediates relevant to molecular evolution, such as the primordial TCA (tricarboxylic acid) cycle. Here, we find evidence that the asteroid Ryugu samples underwent substantial aqueous alteration, as revealed by the presence of malonic acid during keto–enol tautomerism in the dicarboxylic acid profile. The comprehensive data suggest the presence of a series for water-soluble organic molecules in the regolith of Ryugu and evidence of signatures in coevolutionary aqueous alteration between water and organics in this carbonaceous asteroid.

Pristine samples from the near-Earth asteroid (162173) Ryugu returned to Earth by the Hayabusa2 spacecraft provided a valuable opportunity to reveal the organic astrochemistry preserved for over 4.6 billion years in the Solar System[1–4]. This unique opportunity for investigating primordial organic molecules illuminates several scientific contexts involving carbonaceous asteroids, including the following questions[5–7]:

- *What is the role of carbonaceous asteroids in the Solar System history?*
- *What are the origins and characteristics of the light elements, e.g., carbon (C), nitrogen (N), hydrogen (H), oxygen (O), and sulfur (S)?*
- *What do their isotopic compositions reveal?*
- *How do they record the primordial organic evolution on the asteroid?*

- *Is the nature of molecular chirality symmetric or asymmetric?*
- *How do interactions between water, organic matter, and minerals affect chemical diversity?*

To address these important scientific questions, the Hayabusa2 soluble organic matter (SOM) team[6] evaluated aggregate fine grain samples from the first and second touchdown sites (hereafter, TD1 and TD2); hence, the bulk chemistry data from these two sample collections are averaged representative values for the surface (A0106) and possibly subsurface (C0107) environments (i.e., TD2 was near the artificial crater, for which the depth was -1.7 meters below ground level[8]) of Ryugu (Fig. 1). For further insight at the organic molecular level, the SOM team determined the first answers to these questions based on carbon (C), nitrogen (N), hydrogen (H), oxygen

✉e-mail: takano@jamstec.go.jp

Nature Communications | (2024)15:5708                                                                                                    2

(O), sulfur (S) elements and their isotopic profiles[6,9,10], monocarboxylic acids[6], amino acids and their molecular chirality[6,11,12], pyrimidine nucleobase and N-heterocycles[6,9], primordial salts and sulfur-bearing labile molecules between the organic and inorganic interfaces[10], aliphatic hydrocarbons and polycyclic aromatic hydrocarbons (PAHs)[13,14], comprehensive organic molecular profiles[6,15], molecular growth signatures[16], and sub-mm scale spatial imaging for organic homogeneity and heterogeneity in the mineral assemblage[6,17]. According to

Fourier transform-ion cyclotron resonance mass spectrometry (FT-ICR/MS) analysis, the SOM from Ryugu samples contained highly diverse organic molecules (~20,000 species) in the solvent extracts[6,15].

Naraoka et al.[6] reported organic molecular diversity from initial bulk (IB) to insoluble organic matter (IOM) in a sequential extraction process using hydrophilic to hydrophobic solvents. In this report, we determine the molecular diversity of polar organic molecules extracted from the first contact between hot water and pristine Ryugu samples

**Fig. 1 | Profiles of samples obtained from asteroid (162173) Ryugu and various observation photographs from kilometer to micrometer scales. A** Ryugu photograph taken with the Optical Navigation Camera Telescopic (ONC-T). The photo was taken on August 31, 2018. Credit: JAXA, Univ. Tokyo etc. **B** Thermal image of Ryugu from the thermal infrared imager (TIR). The observation indicates that the lowest temperature in the blue section is estimated to be below −50 °C, whereas the lowest temperature in the red section is estimated to be < 60 °C. Please see the onsite data acquisition and temperature dynamics[74,75]. Credit: JAXA etc. The data were collected on August 31, 2018. **C** The surface of the asteroid Ryugu and the shadow of the Hayabusa2 spacecraft. The image was taken from ONC-W1 at an altitude of 70 m. Date taken: 21 September 2018. **D** The 1st touchdown operation on Ryugu with CAM-H imaging on 22 February 2019. The image was captured just before touchdown during descent at an altitude of approximately 4.1 m. Credit: JAXA. **E** Photograph of initial sample A0106 (38.4 mg)[6] from the asteroid Ryugu during the 1st touchdown sampling[1,2]. A photograph of C0107 (37.5 mg) from the 2nd touchdown sampling is shown in Supplementary Fig. S1. The scale bar represents 1 mm (red line). **F** Reference photograph of the discolored and altered cross-section of the Ryugu sample showing possible precipitates (e.g., C0041)[76]. **G** The isotopic compositions of C, N, H, and S of the Ryugu aggregate samples for A0106 and C0107 are shown after compilation[6,9,10]. The isotopic compositions of $\delta^{13}C$ (‰ vs. VPDB), $\delta^{15}N$ (‰ vs. Air), $\delta D$ (‰ vs. VSMOW) and $\delta^{34}S$ (‰ vs. VCDT) are expressed as international standard scales. By comparing the classification of carbonaceous meteorites in the Solar System, the compiled data suggested that the Ryugu sample has isotopic characteristics most similar to the petrologic type of CI chondrite[6,52].

and report the unique color characteristics of the sequentially extracted fractions with systematic variations in their $^{13}C$- and $^{15}N$-isotopic profiles. If indigenous water–organic interactions occurred in the history of the asteroid, the signatures of parent body aqueous alteration could have been recorded in these hydrophilic organic molecules (Fig. 2).

To decipher the chemical evolution that occurred in surface and subsurface samples[1,2,18], we comprehensively evaluated highly diverse hydrophilic organic molecules using capillary electrophoresis (CE) with high-resolution mass spectrometry (HRMS). We used this molecular information to interpret the aqueous alteration processes that asteroid Ryugu has experienced to complement the study by Naraoka et al., who reported organic molecular diversity from initial bulk (IB) to insoluble organic matter (IOM) in the sequential extraction process.

## Results and discussion
### Identification of water-extractable molecules and diverse structural isomers

The Ryugu A0106 and C0107 samples (~10 mg each) were subjected to hot water extraction in sealed ampoules at 105 °C for 20 h for the present study[6] (see Methods). This extraction targeting water-extractable compounds followed previous reports (e.g., hydroxy acids[19,20];). We first identified highly diverse hydroxy acids and hydrophilic molecular groups in hot water extracts by CE-HRMS (Fig. 2). Figure 3A shows the baseline resolution of representative hydroxy acids and other molecules from the hot water extracts identified with reference standards (Murchison meteorite; Methods). We determined each molecule by migration time (MT) and the exact mass corresponding to the monoisotopic mass[9]. Short-chain hydroxy acids (e.g., glycolic acid, $HO\text{-}CH_2\text{-}COOH$; lactic acid, $CH_3\text{-}CH(OH)\text{-}COOH$; and glyceric acid, $HO\text{-}CH_2\text{-}CH(OH)\text{-}COOH$) were predominant in aggregate samples of A0106 and C0107 from Ryugu (Fig. 3B).

Within the concentration range of 10 ppb to $10^3$ ppb [i.e., parts per billion (ppb) as nanograms (ng) hydroxy acid per gram (g) of extracted Ryugu sample] (Table S1), structural isomers of hydroxy acids and molecular abundance were determined. The concentration of lactic acid ($C_3$), which is more abundant than glycolic acid ($C_2$), is consistent with previous reports on the Murchison meteorite[19,20]. Among these homologs of hydroxy acids, we also identified molecules potentially relevant to chemical evolution (e.g., pyruvic acid, $C_3H_4O_3$; mevalonic acid, $C_6H_{12}O_4$; and citric acid, $C_6H_8O_7$). Since these molecules are important precursors in diverse molecular evolution[21], demonstrating their presence on the carbonaceous asteroid Ryugu is significant. Specifically, these molecules are biochemically crucial and are intermediate substrates of the lipid synthesis pathway and Krebs cycle. Chemically reactive hydroxy acids (e.g., glycolic acid) may play an important role in molecular evolution for the formation of primary carbon chains[22]. Furthermore, there may be a connection pathway between hydroxy acids and formose reaction-derived IOM[23] as side products[24].

In addition to the previously reported organic acids (e.g., formic acid and acetic acid[6]) and nitrogen heterocycles[9], we also identified a new group of diverse carboxylic acids (i.e., monocarboxylic acids for aliphatic, aromatic, unsaturated, and keto acids; Figs. 2, 3 and Tables S1, S2) and nitrogen (N)-bearing molecules, including amines (e.g., urea, $CH_4N_2O$; and glycocyamine, $C_3H_7N_3O_2$), hydroxy- and N-heterocyclic indoles (e.g., dihydroxyindole, $C_8H_7NO_2$; and hydroxyindole, $C_8H_7NO$), in hot water extracts. Thus, we suggest that the spectroscopic signals of hydroxyl groups (-OH) and amino/imino groups (-NH) in the infrared spectra (chambers A and C[2]; A0106 and C0107[9]; grain-scale and surface observation; Fig. S13, cf.[17,25]:) include a substantial amount of intramolecular -OH and -NH moieties originating from the series of polar organic molecules in the present study.

### Aqueous alteration signatures and keto–enol tautomerism

Aliphatic dicarboxylic acids (e.g., $C_2$, oxalic acid; $C_3$, malonic acid; $C_4$, succinic acid; $C_5$, glutamic acid; and $C_6$, adipic acid) are defined as organic compounds bearing two carboxyl groups (-COOH) with an aliphatic backbone. We detected dicarboxylic acids (e.g., oxalic acid, malonic acid, succinic acid, glutaric acid, adipic acid, malic acid, and maleic acid) within the concentration range of 10 ppb to $10^3$ ppb (Table S1; Fig. 4A). Previous reports have suggested that the relative concentration of malonic acid ($HOOC\text{-}CH_2\text{-}COOH$) in the dicarboxylic acid group is sensitive properties by the process of keto–enol tautomerization[26,27]. Laboratory-based malonic acid formation has been compared with the extraterrestrial origin of dicarboxylic acids from tautomerization[28]. Enol malonic acid is presumed to decompose faster than other dicarboxylic acids because it produces a thermodynamically unstable carbon–carbon double bond (i.e., $HO\text{-}C = CH\text{-}$, vinyl alcohol group[29–31]) during aqueous alteration as follows:

$$HOOC-CH_2-COOH \text{ with } H_2O \rightarrow \left[(HO)_2-C=CH-COOH\right] \Longleftrightarrow \left[HOOC-CH=C-(OH)_2\right]$$

Hence, the formation of two vinyl alcohol groups on the intramolecular malonic acid is probably more reactive (chemically unstable) than that of other dicarboxylic acids (Fig. 4A, B). After unstable equilibrium is eventually reached under aqueous conditions at higher temperatures[32,33], keto–enol tautomerism induces decarboxylation to form acetic acid ($CH_3COOH$) and carbon dioxide ($CO_2$) as end products (Fig. 4B). Hence, a substantial concentration of acetic acid[6] can result from chemical cleavage of the secondary acetogenic process via malonic acid. Therefore, we suggest that malonic acid (mole%) is a molecular signature of the aqueous alteration process recorded in the asteroid Ryugu. In fact, the relative abundance of malonic acid is an order of magnitude lower than that of CM meteorites (e.g., Murchison and Murray, as shown in Fig. 4A), suggesting a different aqueous history.

### The systematics of hydrophilic molecules at two sampling locations on Ryugu

The systematics for elemental and organic chemical surveys, including CNHOS and hydrophilic molecular groups, were compiled to formulate the TD1 and TD2 diagrams (Fig. 5). Within these overviews of surface and potential subsurface sample profiles[1,2,18], we evaluated

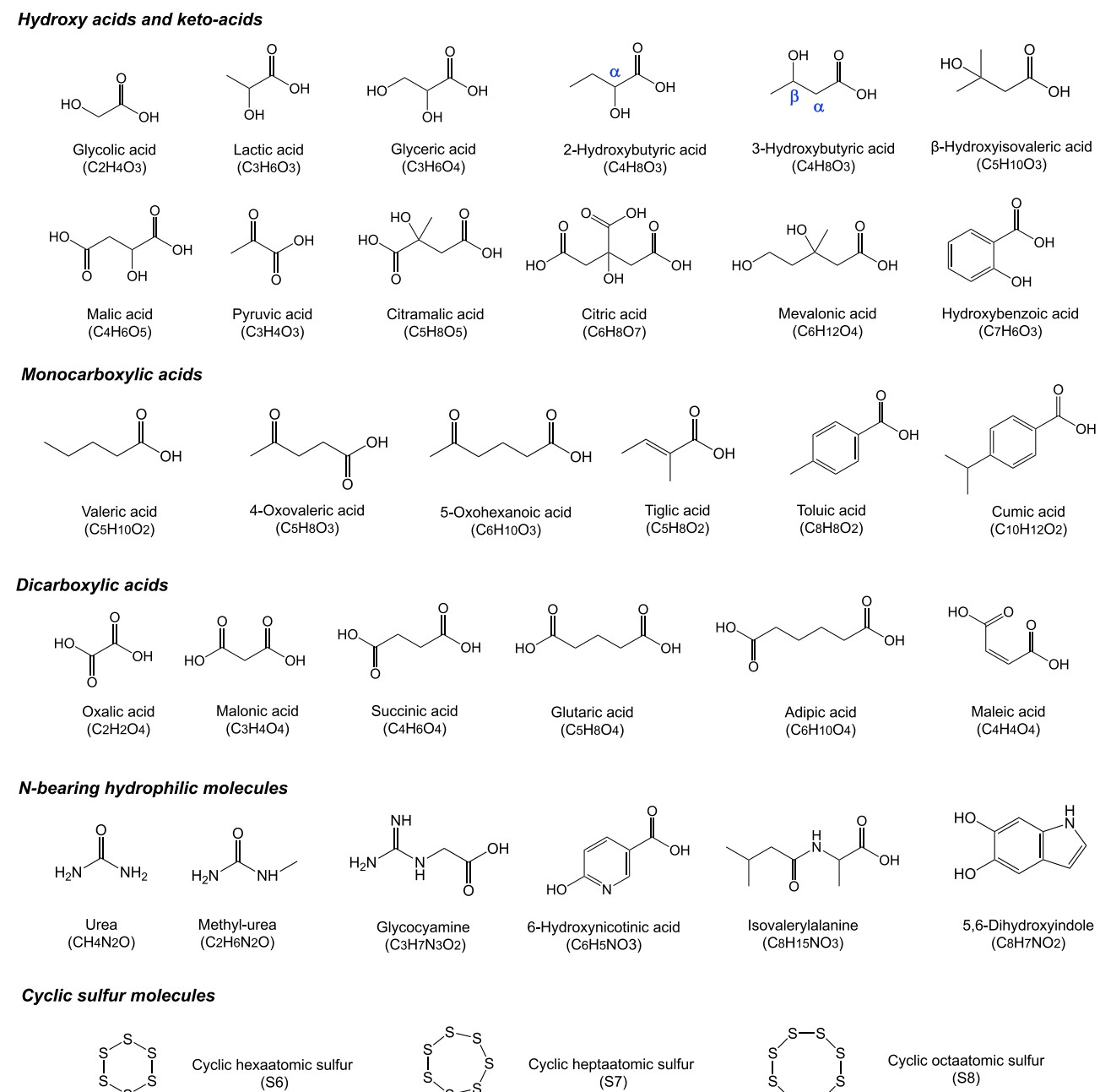

**Fig. 2 | Representative molecular structures of newly identified from Ryugu aggregate samples (A0106 and C0107).** The hot water-extractable molecular structures include α-hydroxy acids (e.g., glycolic acid, lactic acid, and 2-hydroxybutyric acid), β-hydroxy acids (e.g., glyceric acid, 3-hydroxybutyric acid, mevalonic acid, and hydroxybenzoic acid), dicarboxylic hydroxy acids (e.g., malic acid and citramalic acid), monocarboxylic acids (e.g., valeric acid, 4-oxovaleric acid, 5-oxohexanoic acid, tiglic acid, toluic acid, and cumic acid), dicarboxylic acids (e.g., oxalic acid, malonic acid, succinic acid, glutaric acid, adipic acid, and maleic acid), tricarboxylic acid (e.g., citric acid), pyruvic acid and other nitrogen-bearing hydrophilic molecules (e.g., urea, methylurea, glycocyamine = guanidinoacetic acid, 6-hydroxynicotinic acid, isovalerylalanine, and dihydroxyindole). Notably, some hydroxy acids and carboxylic acids have chiral centers with left–right symmetry, but those enantiomers are not discussed in the present report. Newly identified cyclic sulfur compounds ($S_6$, $S_7$ in this study; Supplementary information) were also noted with the comparison of cyclic $S_8$ molecule[6].

the average chemical composition and diversity of hydrophilic molecules to determine whether there is potential organic heterogeneity or homogeneity in Ryugu. The total amount of CNHOS light elements (ΣCNHOS) in the IB of A0106 and C0107 were ~21.3 wt%[6] and ~23.7 wt%[9], respectively (Fig. 5A). Then, ΣCNHOS in the IOM increased by an order of magnitude (Fig. 5B) because the inorganic matrix was eliminated (cf. IOM description[34]).

The overall observations were plotted directly on or near the 1:1 line for hydroxy acids and other hydrophilic molecules (Fig. 5C), water-extractable amino acids and amines for the CHNO molecular series[6,11] (Fig. 5D), and inorganic cations and anions[10] (Fig. 5E). The detection of N-bearing primary amine molecules ($R-NH_2$), ammonium ions ($NH_4^+$)[6,10,11] and urea molecules [$(NH_2)_2 = CO$] (Fig. 5F) from Ryugu is an important finding, not only as evidence of exogenous nitrogen carriers but also as the most primitive chemical forms of nitrogen[35,36]. Urea and alkyl-urea groups (e.g., methyl-urea and alkyl-urea up to $C_6$) may also serve as reservoirs of involatile C, N, O, and H on the asteroid. Urea is also an interesting organic reactive substrate that exhibits amphiphilic

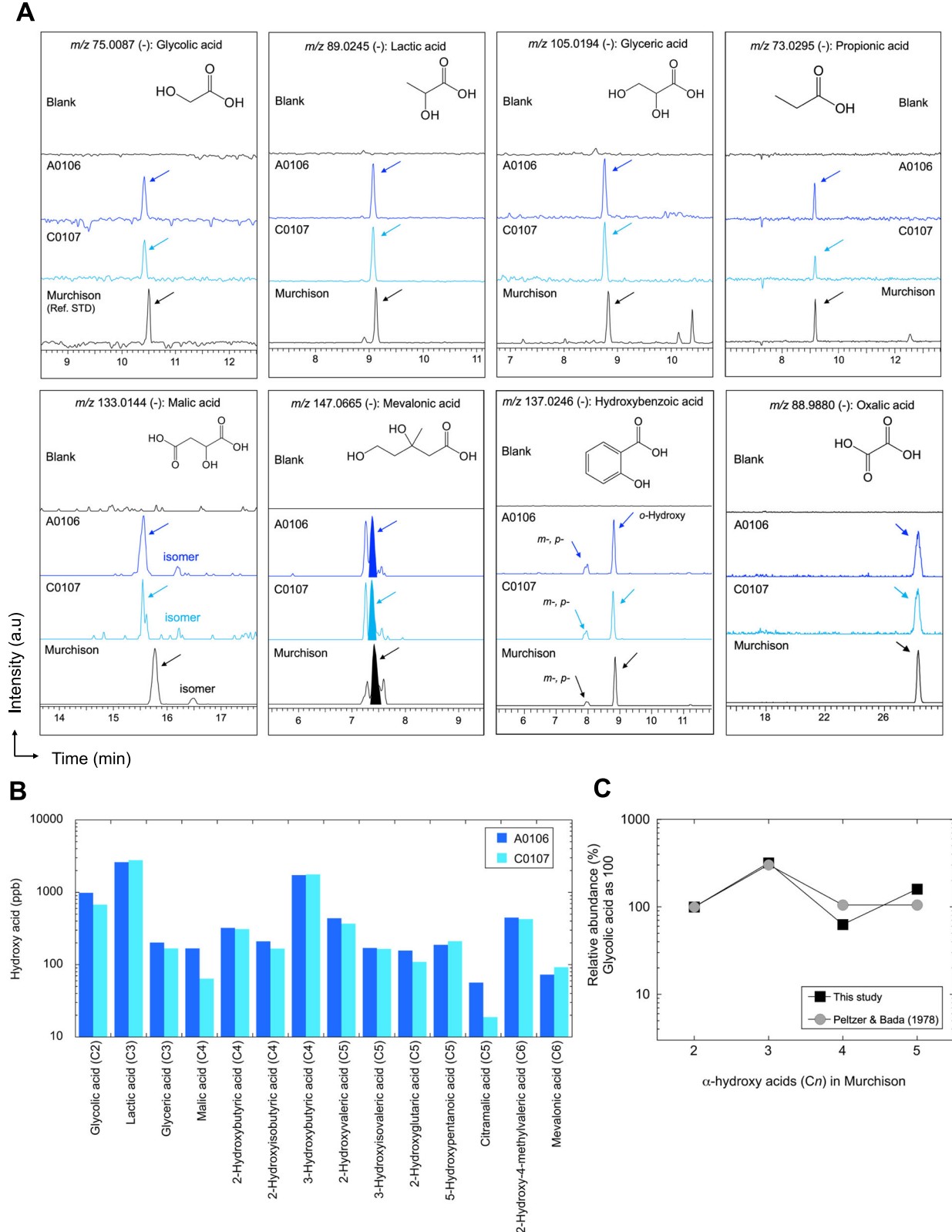

**Fig. 3 | Representative hydrophilic molecular groups in hydroxy acid, dicarboxylic acid, and tricarboxylic acid in hot water extracts from Ryugu samples (A0106 and C0107) and a reference sample (Murchison). A** High-resolution mass electropherogram of capillary electrophoresis during the analysis of hot water extracts (#7-1). The blank was composed of ultrapure water before hot water extraction. Based on the migration time (min) and mass accuracy within -1 ppm (μg/g) of the theoretical peak (*m/z*), we assigned each observed peak to the corresponding standard (Fig. S4). **B** Concentrations of representative hydroxy acids

determined in Ryugu aggregate samples. In this graph, dark blue and light blue represent samples A0106 and C0107, respectively. These hydroxy acids and other related hydrophilic molecules from fraction #7-1 (hot water extracts) are in ppb. **C** The analytical accuracy for the concentration of short-chain α-hydroxy acids (i.e., glycolic acid, lactic acid, 2-hydroxybutyric acid, and 2-hydroxyvaleric acid) extracted from Murchison and Murray meteorites[19] (glycolic acid as 100%) is shown for the same formulation.

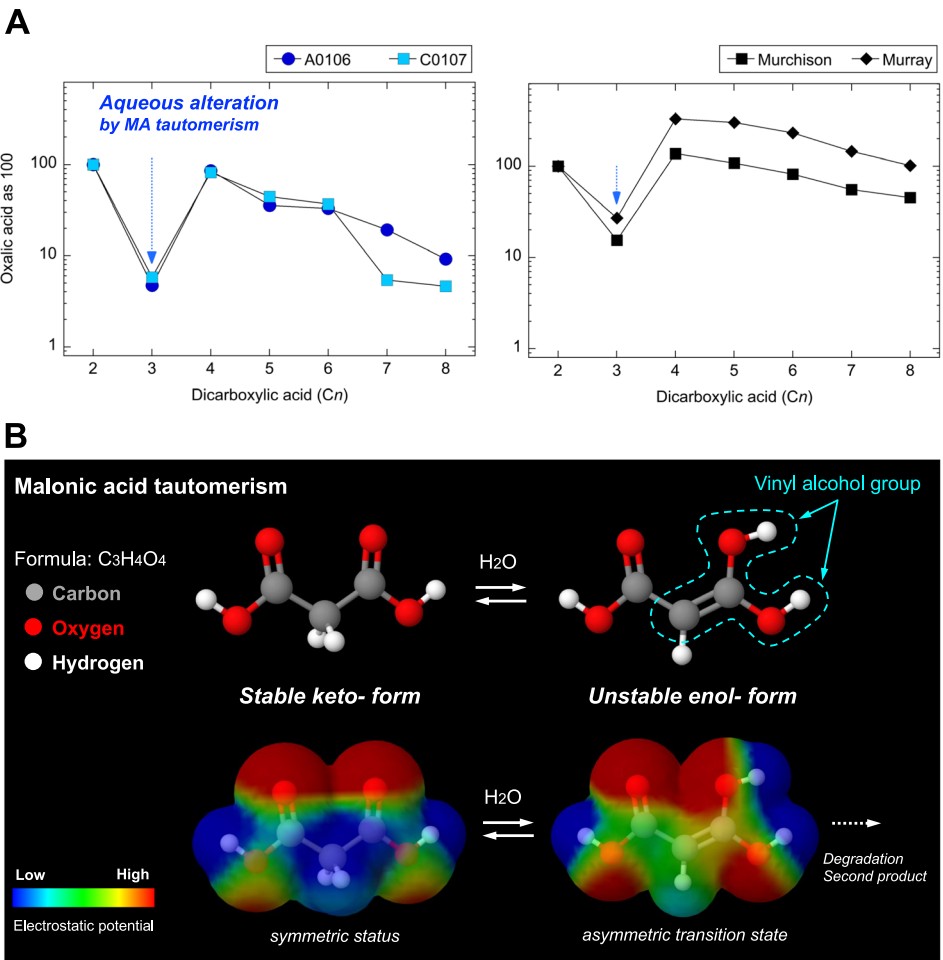

**Fig. 4 | Evidence for aqueous alteration of the asteroid Ryugu revealed by dicarboxylic acids and molecular tautomerism of malonic acid. A** Dicarboxylic acid profiles (i.e., $C_2$, oxalic acid; $C_3$, malonic acid; $C_4$, succinic acid; $C_5$, glutaric acid; $C_6$, aspartic acid; $C_7$, pimelic acid; and $C_8$, suberic acid) for the Ryugu (A0106 and C0107) and CM types (Murchison and Murray) normalized by oxalic acid as 100%. **B** Mechanism underlying keto–enol tautomerism of malonic acid (MA), which converts a chemically stable keto form to an unstable enol- form in the aqueous alteration process. The two enol- forms of the unstable MA tautomer are symmetric and in fact identical molecule.

properties and behaves as a solid and/or liquid depending on temperature and ambient physicochemical factors[22,37]. Regarding the temperature constraint of Ryugu, Yokoyama et al. reported that samples from TD1 and TD2 remained below -100 °C after aqueous alteration until the present based on the abundance of structural water[38].

To further describe the CI-like organic characteristics, the hydrophilic molecules from Ryugu (A0106 and C0107) were compared to CM-type chondrites from Murchison and Murray (Fig. 6A). According to the composition of amino acids found in the CI-type meteorite Ivuna[39], the properties of meteoritic amino acids were verified for Ryugu with the same normalization (Fig. 6B). Compared to the CM2 chondrites of Murchison and Murray, CI-type carbonaceous chondrites with parent bodies that have experienced aqueous alteration contain lower total amino acid abundances[39,40]. In this context, Burton and coworkers reported that carbonaceous chondrites that experienced high-temperature thermal alteration along with aqueous alteration (e.g., CI type Y-980115; re-examination with $\delta^{15}N$ of amino acids[41]) have much lower amino acid abundances than CI Orgueil and CM Murchison meteorites[40,42]. Distinct positive correlations were observed in both concentration profiles above the 1:1 line, whereas the principal component-2 (PC2) scores suggested that the concentration of hydrophilic molecules was lower and that the history of aqueous alteration differed between the Ryugu and CM samples (Fig. 6C). Therefore, we suggest that comprehensive surveys of meteoritic amino acids of the CI and CM types are important for classifying Ryugu[6,11,13].

## Stepwise $^{15}N$ depletion and $^{13}C$ depletion during solvent extractions

The mass balance equation[1] for the initial bulk composition of organic matter (normalized to 100% for IB as whole rock) in the Ryugu sample is expressed as the sum of inorganic fractions[10], soluble and insoluble organic fractions through the following equation:

$$IB = \Sigma\,Inorganics + \Sigma\,SOM + \Sigma\,IOM \qquad (1)$$

$\Sigma SOM$ represents the sum of the components extracted in each process of sequential extraction, whereas $\Sigma IOM$ represents the sum of the insoluble organic fractions, as detailed in previous literature[6,34]. We investigated the nitrogen isotopic profiles during sequential solvent extraction by hot water extracts (#7-1), formic acid extracts (#9), and HCl extracts (#10) for Ryugu (A0106 and C0107) and the CI group reference (Orgueil meteorite[9,10]) (Fig. 7A). Interestingly, this validation clearly showed that organic solvent extraction resulted in $^{15}N$-enriched profiles (e.g., hot water extracts; < +63.1‰ and < +55.2 ‰ vs. Earth's atmospheric air for A0106 and C0107, respectively) for each extractable organic fraction during the sequential process. Therefore, the nitrogen isotopic composition of the insoluble residue indicated that it

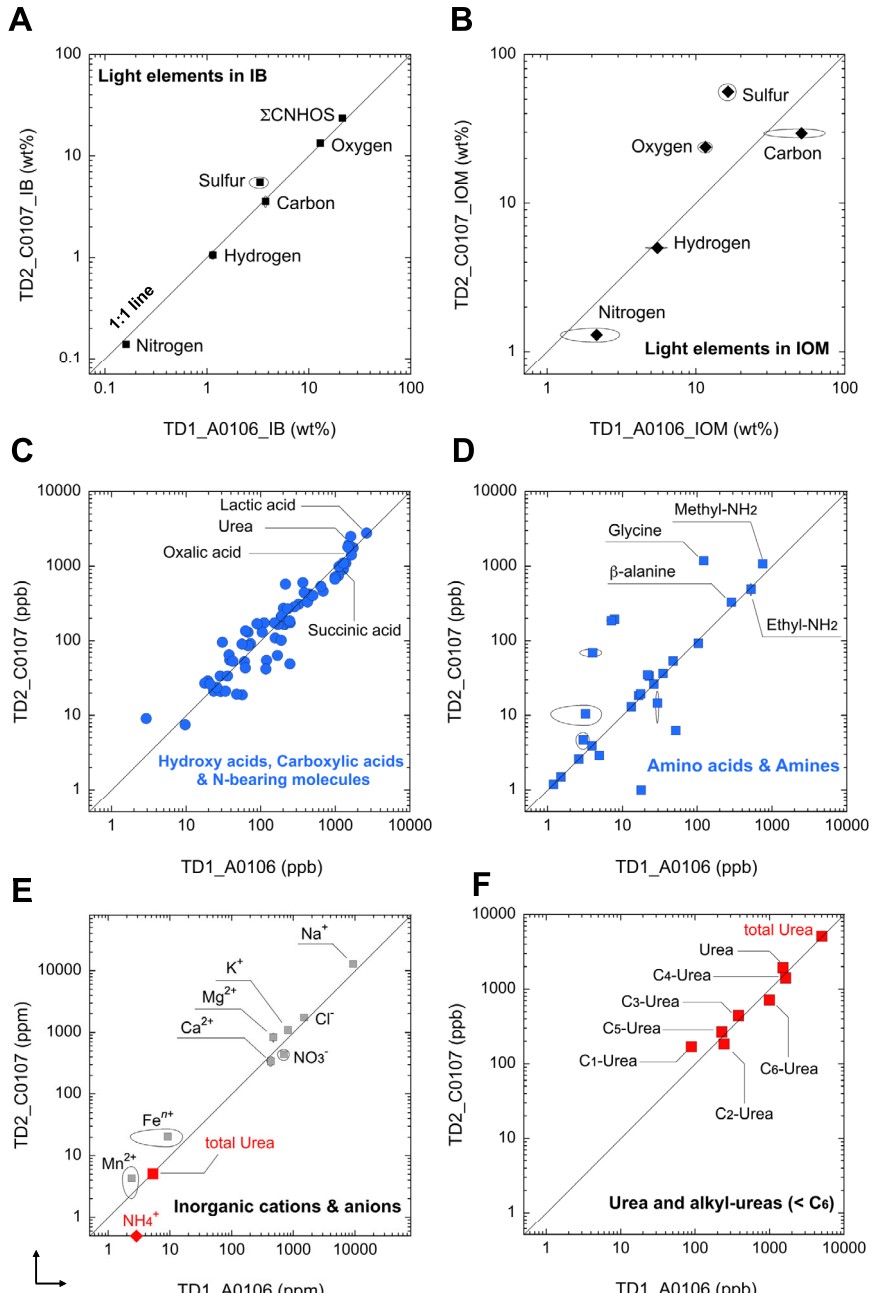

**Fig. 5 | Standardization to comparatively verify the elemental and hot water-extractable molecular properties of samples from the 1st touchdown site (TD1) and 2nd touchdown site (TD2) at Ryugu.** Notably, the Ryugu sample is of scientific value as a surface (TD1) and subsurface sample (TD2) from the carbonaceous asteroid[1,18]. In this report, we evaluated the hydrophilic organic molecules in surface aggregate (A0106) and subsurface aggregate (C0107) samples as follows. **A** Light elements in IB samples for total C, N, H, and S and pyrolyzable O in wt%. Compilation after the references[6,9,10]. The error arc indicates the standard deviation (1σ). Here, we define IB as whole-rock bulk, which includes all inorganic matrices such as silicates and carbonates, and IOM as the fraction that does not contain silicates[43]. **B** CNHOS contents in IOM (sample treatment[34] and measurement by the present report) in wt% (Table S3). **C** Hydroxy acids, carboxylic acids and other newly identified N-bearing hydrophilic molecules in this study obtained from fraction #7-1 (hot water extracts) in ppb. The molecular assignments and raw data profiles are shown in Fig. 3 and Tables S1, S2, respectively. **D** Amino acids and amines from fraction #7-1 (hot water extraction) in ppb. The data were compiled after the references[6,11]. Please see the error notation in the diagram[11]. **E** Major inorganic cations and anions from fraction #7-1 (hot water extracts) on the ppm scale. Please see the report for ammonium ion detection ($NH_4^+$, ~ 3 ppm; red diamond symbol) and other important molecules associated with organic and inorganic profiles[10]. The error notations in the diagram indicate 2σ after the reference. **F** Concentrations of urea and alkyl-urea (i.e., methyl-urea, ethyl-urea, and other alkyl ureas up to $C_6$-urea) were measured in the present study.

was conversely depleted of [15]N-organic matter in the stepwise extraction (Fig. 7B). We observed that the carbon isotopic composition of the insoluble residues also tended to be [13]C-depleted down to −17.0 ± 0.2 ‰, as observed for [15]N profiles (down to +28.2 ± 3.8 ‰). This observation (Fig. 7B) agrees well with previous reports on the carbon and nitrogen isotopic compositions of extractable SOM and refractory

IOM in Murchison[43]. In contrast, it is interesting to note that the sulfur isotopic composition ($\delta^{34}S$) converged to the VCDT scale (-0‰) before and after solvent extraction. Within the SOM fraction, the normalized nitrogen balance of each extract was high in the formic acid fraction, indicating that the pink extracts (A0106 and C0107) contained a substantial amount of hydrophilic organic matter (Supplementary

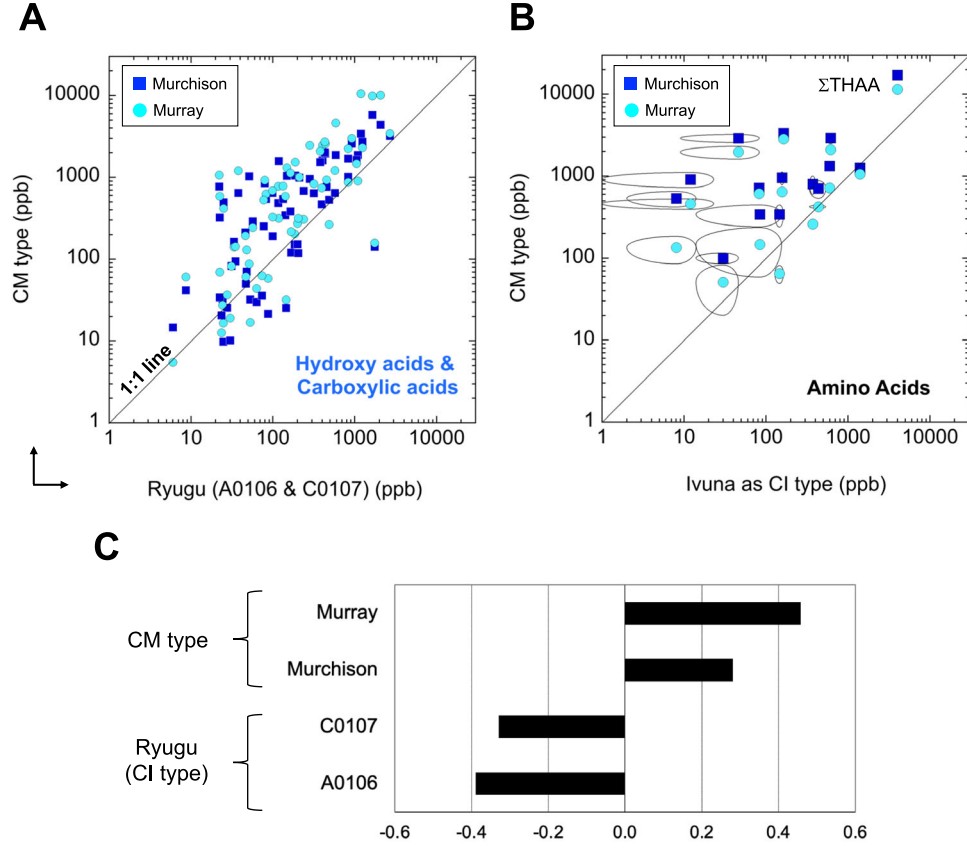

**Fig. 6 | Summary of integrated observations of Ryugu with CI type (Ivuna) and CM type (Murchison and Murray) for aqueous alteration processes throughout their history. A** Hydroxy acids, dicarboxylic and tricarboxylic acids, and other newly identified hydrophilic molecules for comparison between Ryugu (this study) and CM (Murchison and Murray; this study) type at the ppb scale. The Ryugu values on the horizontal axis are shown as the average of A0106 and C0107 (Table S1). **B** Amino acids for the comparison between CI type (Ivuna) and CM type (Murchison and Murray) based on compilation[39]. Please see the individual molecular information in the diagram with the following review[40] and amino acid profiles for Ivuna and Orgueil (Fig. S14). The error arc indicates the standard deviation ($1\sigma$). **C** Principal component (PC) analysis between Ryugu and CM (Murchison and Murray) regarding hydroxy acids and dicarboxylic and tricarboxylic acids with other hydrophilic molecules based on the panel (A) raw data profile (this study). The PC2 scores between the CM type (Murchison, Murray for sample description[77]) and CI type of Ryugu (A0106 and C0107; Tables S1, S2) are shown, suggesting a different history of indigenous organic molecules.

Information). Based on the present observations (Fig. 7), the hypothesis regarding isotope fractionation during the formation of meteoritic organic matter[44], volatile nitrogen molecules and thermally altered N residues in Ryugu[45,46], and primordial $^{15}N$ depletion in the protosolar nebula (down to −400‰)[36] will be important for describing nitrogen dynamics in the Solar System.

**Implication of aqueous alteration history on the parent body**
When investigating the history of the carbonaceous asteroid (162173) Ryugu, we found definite signatures of aqueous alteration from hydrophilic organic molecules, as shown in the hypothetical concept summary (Fig. 8). We consider that physicochemical and temperature factors (i.e., cold and hot thermal conditions; and icy dry and aqueous wet cycles, Fig. 1B) correlate with the molecular evolution between water and organic matter within the cold hydrothermalism[15]. The coevolutionary outline hypothesized here is also supported by the observations of secondary mineral assemblages and altered vein formations[38,47–49] (Fig. 9). For a comparative investigation of those findings, the origin of Ryugu's water within the history of the parent body will be elucidated in subsequent studies[38,50–52]. As a notable opportunity in 2023, NASA's OSIRIS-REx (Origins, Spectral Interpretation, Resource Identification and Security-Regolith Explorer) spacecraft returned the carbonaceous asteroid (101955) Bennu sample

to Earth[53]. We expect that international return missions will offer extremely important scientific opportunities to explore the history of organic chemical evolution.

We hope that the Bennu sample will reveal detailed information on chemical evolution and molecular chirality[6,11,40,54], including widely diverse hydrophilic molecules in the asteroid history. Notably, the carbonate veins observed on some boulders at Bennu[55] are unique and should reveal interactions between pristine aqueous alteration processes, as discussed in this paper and other perspectives[7,53,56]. Therefore, we conclude here that carbonaceous asteroids are natural laboratories for observing realistic primordial molecular evolution in organic and inorganic contexts.

## Methods

**Sample process and extraction of hydrophilic organic molecules**
The description summary of the onsite sample collection from the asteroid Ryugu is reported by the Hayabusa2 International Team[1,2]. To ensure the quality of the pristine sample, the project team performed an environmental evaluation in the prelaunch phase[57,58], system design and preliminary assessments[59,60] and careful assessment of the sample process during volatile recovery in Australia[61,62] until the curation facility[63,64]. The seamless sample process and the extraction of organic molecules from Ryugu have been described previously[6]

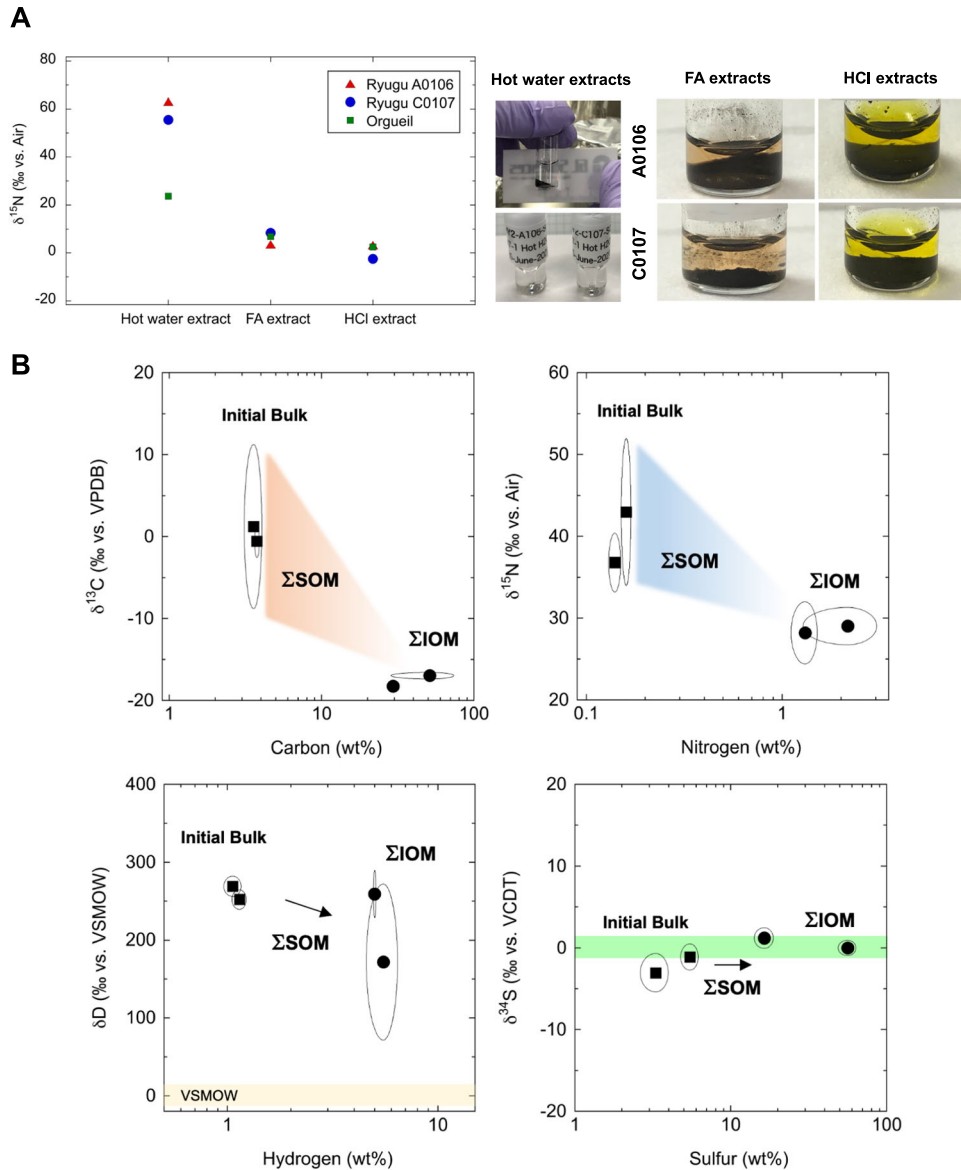

**Fig. 7 | Carbon, nitrogen, hydrogen and sulfur abundances and their isotopic profiles before and after the solvent extraction processes from the organic matter facies. A** The ¹⁵N-nitrogen isotopic depletion between the supernatant and IOM residue during sequential solvent extraction for the Ryugu (A0106 and C0107) and Orgueil samples. The pinkish color originates from formic acid extract #9, and the yellowish color originates from HCl extract #10. The other chemical profiles are shown in Figs. S6, S7, and S8. Please also see the residue of IOM (black color) on the bottom of the vial[78]. Unique brownish colloidal-colored fractions (#4 MeOH extract, #5 water extract) were observed for A0106 and C0107 (cf. Figs. S5, S9).

**B** Carbon, nitrogen, hydrogen, and sulfur profiles (wt%) and their isotopic shifts observed from IB, ΣSOM and ΣIOM. The data were compared with previous references[6,9,10] and this study. The abundance of carbon (wt%), nitrogen (wt%), hydrogen (wt%) and sulfur (wt%) in IOM increased by one order of magnitude because of the dissolution of silicates and other mineral structures. The error arc indicates the standard deviation (1σ). Note the previous reports regarding volatile components[45,46] and inorganic profiles[10,38]. The isotopic profiles of ΣSOM from the sequential extractions are shown in Table S4. Please see the IOM treatment[34] and C-N-S isotopic variations of the Solar System[6,36,79,80].

(Supplementary Information, Figs. S1–S3). The extracted fractions were photographed (this study; Figs. S5–S8) and analyzed by the SOM team. The insoluble organic residue was processed by the IOM team[34].

### Analysis of hydrophilic molecules for hydroxy acids and mono-, di-, tri-carboxylic acids

We performed capillary electrophoresis-high-resolution mass spectrometry (CE-HRMS) using the ω Scan package (Human Metabolome Technologies, Inc., Japan) as described in previous reports[9,65]. In brief, CE-HRMS analysis was performed with an Agilent 7100 CE capillary electrophoresis system (Agilent Technologies, Inc., Santa Clara, CA, USA) equipped with a Q Exactive Plus (Thermo Fisher Scientific Inc., Waltham, MA, USA), Agilent 1260 isocratic HPLC pump, Agilent

G1603A CE-MS adapter kit, and Agilent G1607A CE-ESI-MS sprayer kit (Agilent Technologies, Inc., Santa Clara, CA, USA). The system was controlled with Agilent MassHunter workstation software for LC/MS data acquisition for the 6200 series TOF/6500 series Q-TOF version B.08.00 (Agilent Technologies, Inc., Santa Clara, CA, USA) and Xcalibur (Thermo Fisher Scientific Inc., Waltham, MA, USA). The separation was performed with a fused silica capillary (50 μm i.d. × 80 cm total length) and electrophoresis buffer (H3301-1001, HMT) as the electrolyte. To ensure the accuracy of the analysis, blank measurements were also performed to validate the raw data acquisition. Compound peaks were extracted using MasterHands, and automatic integration software was used to obtain raw signal information, including *m/z* values, peak areas, and migration times (MTs)[66].

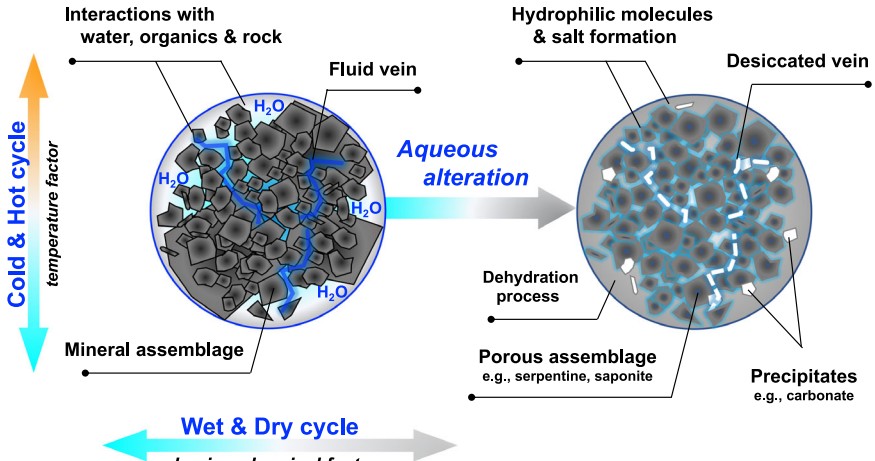

**Fig. 8 | Aqueous alteration of primordial hydrophilic organic molecules and minerals during parent body processing of asteroid (162173) Ryugu.** The left panel represents the initial primary mineral assemblage and fluid veins in the early stage of interaction between water, organics, and rock within the bedrock. The right panel represents altered secondary mineral assemblages (i.e., porous and physically fragile), desiccated veins, and precipitates in the late stage and ongoing stage with dehydration processes at Ryugu[47,50]. Within cold hydrothermalism[15], thermal history in the asteroid[74,75], and temperature constraints[38], this figure conceptualizes aqueous alteration, and the sizes of regolith particles and bedrock are arbitrary scales. Amino acids and other hydrophilic molecules[6] with "salt" formation[10] are overviewed in the illustration diagram of chemical evolution. The organic analysis of the asteroid Bennu[81] is a valuable opportunity to consider the scientific consequences of this study.

We used the most representative carbonaceous meteorite of Murchison[6,9,67] as a reference standard to confirm our qualitative evaluation of the sample matrix effects (Fig. S4). The standard mixture including the working reagents for migration time alignment (e.g., AM1, AM2, AM3, AM4, and AM5) and an internal standard for anion analysis (ISA) were prepared from an HMT metabolomics kit (Human Metabolome Technologies Inc., Tsuruoka, Japan)[65,66,68].

## Tracing CNHSO contents and their isotopic compositions to the IOM fraction

For further isotopic analysis of the organic extracts and IOM residues, we analyzed the elemental abundances of carbon (C, wt%), nitrogen (N, wt%), hydrogen (H, wt%), and sulfur (S, wt%) with isotopic compositions of $\delta^{13}C$ (‰ vs. VPDB), $\delta^{15}N$ (‰ vs. Air), $\delta D$ (‰ vs. VSMOW), and $\delta^{34}S$ (‰ vs. VCDT), respectively[6,9,10] (Fig. S5). For the total CNS contents and their isotopic compositions ($\delta^{13}C$, $\delta^{15}N$, $\delta^{34}S$), we used an ultrasensitive nano-EA/IRMS method (Flash EA1112 elemental analyzer/Conflo III interface/Delta Plus XP isotope ratio mass spectrometer, Thermo Finnigan Co., Bremen) at JAMSTEC[69,70] (within wide isotopic dynamic ranges in Fig. S10). Analytical validations using the nano-EA/IRMS system were performed during practical analyses and studies on carbonaceous chondrites[41,71]. For the total H and the isotopic compositions ($\delta D$), we used a high-sensitive EA/IRMS method (Delta Plus XL isotope ratio mass spectrometer, Thermo Finnigan Co., Bremen) at Kyushu University. The elemental CNH contents (wt%) and their isotopic compositions ($\delta^{13}C-\delta^{15}N-\delta D$ profiles) of Ryugu samples A0106 and C0107 are shown in Fig. 1G based on the compilation[6,9,10]. The $\delta$ values of the Ryugu samples for C, N, H and S isotopic compositions are denoted using international isotope standards as follows:

$$\delta^{13}C = [(^{13}C/^{12}C)_{Ryugu}/(^{13}C/^{12}C)_{VPDB}-1] \times 1000 \, (‰) \quad (2)$$

with the Vienna Pee Dee Belemnite (VPDB) standard;

$$\delta^{15}N = [(^{15}N/^{14}N)_{Ryugu}/(^{15}N/^{14}N)_{Air}-1] \times 1000(‰) \quad (3)$$

with the Earth atmospheric nitrogen (Air) standard;

$$\delta D = [(D/H)_{Ryugu}/(D/H)_{VSMOW} - 1] \times 1000(‰) \quad (4)$$

with the Vienna Standard Mean Ocean Water (VSMOW) standard; and

$$\delta^{34}S = [(^{34}S/^{32}S)_{Ryugu}/(^{34}S/^{32}S)_{VCDT}-1] \times 1000(‰) \quad (5)$$

with the Vienna Canyon Diablo Troilite (VCDT) standard, respectively. Since the IOM fraction comprises the main portion of various solid organic carbon in Ryugu samples, simultaneous data acquisition for SOM and IOM was performed[6,34].

## Surface-assisted laser desorption/ionization mass spectrometry (SALDI-MS)

SALDI-MS has been used to analyze many materials, including carbonaceous meteorites[72,73], at Tohoku University. Briefly, a matrix-assisted laser system (AP-SMALDI5, TransMIT) connected to an orbital trap mass spectrometer (QExactive, Thermo Fisher Scientific Inc., Waltham, MA, USA) was used to acquire SALDI mass spectra. Mass spectrometry was conducted in positive mode with a mass resolution of 140,000 using a solid-state laser of 20 μm, 60 Hz, and 30 pulses for each spot (Fig. S8). Approximately 130 spots in a $300 \times 300$ μm area in the pit were scanned by the laser.

## FTIR spectra and ultraviolet–visible spectra of the organic extracts

We compiled the Fourier transform infrared spectroscopy (FTIR) profiles of the solvent extracts by using a Nicolet iN10 infrared microscope (Thermo Fisher Scientific Inc., Waltham, MA, USA) between A0106 and C0107 (method after the ref. 6). Briefly, 1–2 μL of the solvent extract was dropped onto a BaF₂ plate (1 mm thick) and air-dried (Fig. S9). The data acquisition for transmission spectra was performed by an MCT (mercury–cadmium–telluride) detector at liquid N₂ in a clean room at Kyushu University. The microscope and detector were continuously purged with dry N₂ gas during analysis.

The ultraviolet–visible (UV–vis) spectra of the extracts were analyzed with a microvolume UV–Vis spectrophotometer (NanoDrop One C, Thermo Fisher Scientific Inc., Waltham, MA, USA) in the wavelength range of 190 nm to 1100 nm (Fig. S7). This spectroscopic measurement was performed at Tohoku University.

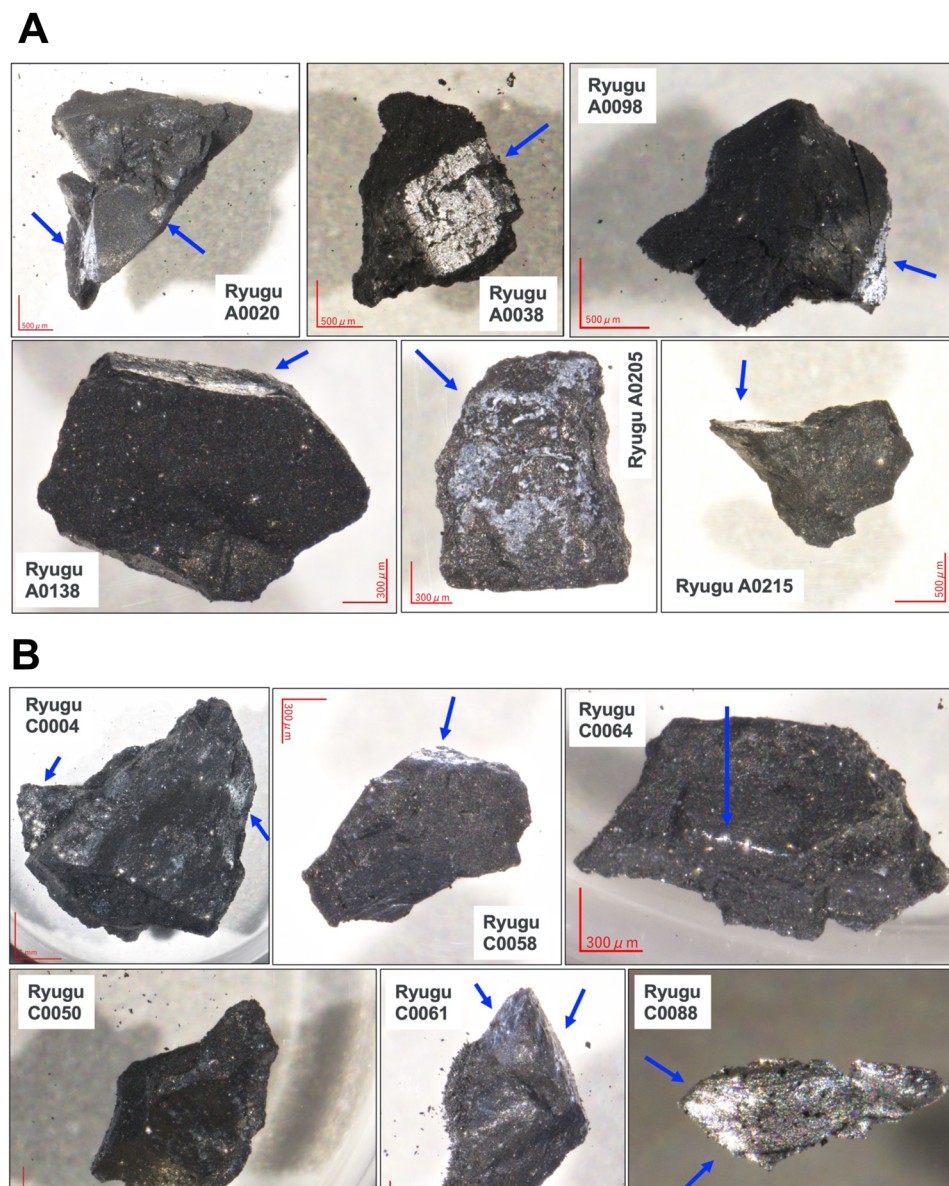

**Fig. 9 | Photographs showing representative altered aqueous signatures, desiccated veins, and spatial cross-sections of Ryugu samples. A** Representative photographs of the chamber A sample series and (**B**) chamber C sample series, showing the cross-sectional areas that are past fluid veins and/or hydrothermally produced precipitates, which are indicated by blue arrows. For a description of the sample properties of organic homogeneity or heterogeneity[17,25], please see the details in a previous report[1,76,82].

## Gas chromatography/mass spectrometry (GC/MS) of hexane extracts from the IOM fraction

After discovering the yellow sticky deposit on the wall in the glass vial containing the IOM fraction (see the pretreatment[34]), we conducted an *n*-hexane extraction to identify cyclic sulfur molecules (i.e., cyclic hexaatomic sulfur, $S_6$; cyclic heptaatomic sulfur, $S_7$; and cyclic octaatomic sulfur, $S_8$) from the fraction (Figs. S11, S12). We analyzed the extracts by gas chromatography/mass spectrometry (GC/MS; 7890B GC and 5975 C MSD, Agilent Technologies, Inc., Santa Clara, CA, USA) with a VF-5MS column (30 m × 0.25 mm i.d., 0.10 µm film thickness, Agilent Technologies, Inc., Santa Clara, CA, USA) at JAMSTEC. The GC oven temperature was programmed as follows: the temperature was initially 40 °C, ramped up at 30 °C min$^{-1}$ to 120 °C, ramped up at 6 °C min$^{-1}$ to 320 °C, and maintained for 20 min. The target molecules were verified by comparison with authentic standards of aliphatic hydrocarbons in *n*-hexane solution (Supplementary Information) and the library database from NIST (National Institute of Standards and Technology).

## Data availability

We declare that all these database publications are compliant with ISAS data policies (www.isas.jaxa.jp/en/researchers/data-policy/). The Hayabusa2 project is releasing raw data on the properties of the asteroid Ryugu from the Hayabusa2 Science Data Archives (DARTS, https://www.darts.isas.jaxa.jp/planet/project/hayabusa2/) for Optical Navigation Camera (ONC), Thermal InfraRed Imager (TIR), Near InfraRed Spectrometer (NIR), LIght Detection And Ranging (LIDAR), SPICE kernels, and PDS4.

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

## Acknowledgements

The Hayabusa2 project has been organized by JAXA (Japan Aerospace Exploration Agency) with DLR (German Space Center), CNES (French Space Center), NASA (National Aeronautics and Space Administration) and ASA (Australian Space Agency). Preliminary results of this study were partly reported at the Hayabusa symposium and Lunar and Planetary Science Conference (LPSC). This study was partly conducted by the official collaboration agreement through the joint research project with JAMSTEC, Keio University and HMT Inc. The authors thank Dr. M. Tomita and Mr. K. Hashizume for their constructive advice and technical cooperation. This research is partly supported by the grant from the Japan Society for the Promotion of Science (JSPS) with KAKENHI

numbers; 21KK0062 (Y.T.), 21J00504 (T.K.), 21H04501&21H05414 (Y.O.), 20H00202 (H.N.). J.P.D. and D.P.G. thank NASA for support of the Consortium for Hayabusa2 Analysis of Organic Solubles. This study was conducted in accordance with the Joint Research Promotion Project at the Institute of Low Temperature Science, Hokkaido University (21G008 and 22G008 to Y.T., Y.O., H.N.).

## Author contributions

Y.Takano, H.N., J.P.D. designed the outline and entire working flow in this study. H.N. and Y.Takano conducted sequential solvent extraction and distributed the SOM samples. Y.Takano, K.Sasaki, H.S. and T.K. conducted the analysis of high-resolution mass spectrometry. N.O.O., Y.Takano, and N.O. conducted the analysis of elemental and isotopic compositions. N.O.O. and N.O. provided the series of authentic C, N isotope standards. H.N. provided the series of H, O isotope standards. T.Yoshimura and Y.Takano lead the primary description of water-extractable cations and anions. Y.O. and T.K. lead the primary surveys of N-heterocycles. E.T.P., K.Hamase and J.C.A. lead the primary surveys of amino acids and amines. D.P.G. and J.P.D. assessed the organic feature between CI and CM type with Ryugu profiles. Y.F., S.M.N., J.Aoki, K.K. performed small-scale analysis of UV spectra and SALDI mass spectrometry. P.S. and F.R.O.D. lead the non-target comprehensive molecular survey and chemical assignments. H.N., Y.Takano, J.P.D. designed the SOM scheme during the initial analysis timelines (~31-May-2022). S.Tachi, H.Yurimoto, T.Nakamura, T.Noguchi, R.O., H.Yabuta, K.Sakamoto lead the initial analysis processes. M.A., T.Yada, M.N., K.Y., A.N., A.M., T.O., and T.U. curated samples. M.Y., T.S., S.Tana, F.T., S.Nakazawa, S.W., and Y.Tsuda contributed to the sample collection at Ryugu. The Hayabusa2-initial-analysis SOM team members are shown in this report. All authors discussed the results, and commented on the manuscript.

## Competing interests

The authors declare no competing interests.

## Additional information

Yoshinori Takano [1,2] ✉, Hiroshi Naraoka [3], Jason P. Dworkin [4], Toshiki Koga [1], Kazunori Sasaki[2,5], Hajime Sato [5], Yasuhiro Oba [6], Nanako O. Ogawa [1], Toshihiro Yoshimura [1], Kenji Hamase[7], Naohiko Ohkouchi[1], Eric T. Parker[4], José C. Aponte [4], Daniel P. Glavin [4], Yoshihiro Furukawa [8], Junken Aoki [9], Kuniyuki Kano [9], Shin-ichiro M. Nomura [10], Francois-Regis Orthous-Daunay[11], Philippe Schmitt-Kopplin[12,13,14], Hayabusa2-initial-analysis SOM team*, Hisayoshi Yurimoto [15], Tomoki Nakamura[8], Takaaki Noguchi [16], Ryuji Okazaki[3], Hikaru Yabuta [17], Kanako Sakamoto [18], Toru Yada[18], Masahiro Nishimura[18], Aiko Nakato [18], Akiko Miyazaki [18], Kasumi Yogata[18], Masanao Abe[18], Tatsuaki Okada [18], Tomohiro Usui[18], Makoto Yoshikawa[18], Takanao Saiki[18], Satoshi Tanaka[18], Fuyuto Terui[19], Satoru Nakazawa [18], Sei-ichiro Watanabe [20], Yuichi Tsuda[18] & Shogo Tachibana [18,21]

[1]Biogeochemistry Research Center (BGC), Japan Agency for Marine-Earth Science and Technology (JAMSTEC), Natsushima, Yokosuka 237-0061, Japan. [2]Institute for Advanced Biosciences (IAB), Keio University, Kakuganji, Tsuruoka, Yamagata 997-0052, Japan. [3]Department of Earth and Planetary Sciences, Kyushu University, 744 Motooka, Nishi-ku, Fukuoka 819-0395, Japan. [4]Solar System Exploration Division, NASA Goddard Space Flight Center, Greenbelt, MD 20771, USA. [5]Human Metabolome Technologies Inc., Kakuganji, Tsuruoka, Yamagata 997-0052, Japan. [6]Institute of Low Temperature Science (ILTS), Hokkaido University, N19W8 Kita-ku, Sapporo 060-0819, Japan. [7]Graduate School of Pharmaceutical Sciences, Kyushu University, Fukuoka 812-0054, Japan. [8]Department of Earth Material Science, Tohoku University, Sendai 980-8578, Japan. [9]Department of Health Chemistry, Graduate School of Pharmaceutical Sciences, The University of Tokyo, Hongo, Tokyo 113-0033, Japan. [10]Department of Robotics Graduate school of Engineering, Tohoku University, Sendai 980-8579, Japan. [11]Université Grenoble Alpes, Centre National de la Recherche Scientifique (CNRS), Centre National d'Etudes Spatiales, L'Institut de Planétologie et d'Astrophysique de Grenoble, 38000 Grenoble, France. [12]Technische Universität München, Analytische Lebensmittel Chemie, 85354 Freising, Germany. [13]Max Planck Institute for Extraterrestrial Physics, 85748 Garching bei München, Germany. [14]Center for Research and Exploration in Space Science and Technology, NASA Goddard Space Flight Center, Greenbelt, MD 20771, USA. [15]Department of Earth and Planetary Sciences, Hokkaido University, Sapporo 060-0810, Japan. [16]Department of Earth and Planetary Sciences, Kyoto University, Kyoto 606-8502, Japan. [17]Department of Earth and Planetary Sciences, Hiroshima University, Higashi-Hiroshima 739-8526, Japan. [18]Institute of Space and Astronautical Science (ISAS), Japan Aerospace Exploration Agency (JAXA), Sagamihara 252-5210, Japan. [19]Kanagawa Institute of Technology, Atsugi 243-0292, Japan. [20]Department of Earth and Environment Sciences, Nagoya University, Nagoya 464-8601, Japan. [21]UTokyo Organization for Planetary and Space Science (UTOPS), University of Tokyo, 7-3-1 Hongo, Tokyo 113-0033, Japan. *A list of authors and their affiliations appears at the end of the paper. ✉e-mail: takano@jamstec.go.jp

## Hayabusa2-initial-analysis SOM team

Hiroshi Naraoka [3], Yoshinori Takano [1,2] ✉, Jason P. Dworkin [4], Kenji Hamase[7], Aogu Furusho[7], Minako Hashiguchi[20], Kazuhiko Fukushima[20], Dan Aoki[20], José C. Aponte [4], Eric T. Parker[4], Daniel P. Glavin [4], Hannah L. McLain[4,14,22], Jamie E. Elsila[4], Heather V. Graham[4], John M. Eiler[23], Philippe Schmitt-Kopplin[12,13,14], Norbert Hertkorn[12], Alexander Ruf[21], Francois-Regis Orthous-Daunay[11], Cédric Wolters[11], Junko Isa[24,25], Véronique Vuitton[11], Roland Thissen[26], Nanako O. Ogawa [1], Saburo Sakai[1], Toshihiro Yoshimura [1], Toshiki Koga [1], Haruna Sugahara[18], Naohiko Ohkouchi[1], Hajime Mita[27], Yoshihiro Furukawa [8], Yasuhiro Oba [6], Yoshito Chikaraishi[6], Takaaki Yoshikawa[28], Satoru Tanaka[29], Mayu Morita[30], Morihiko Onose[30], Daisuke Araoka[31], Fumie Kabashima[32], Kosuke Fujishima[2,24], Hajime Sato[2,5], Kazunori Sasaki[2,5], Kuniyuki Kano [9], Shin-ichiro M. Nomura [10], Junken Aoki [9], Tomoya Yamazaki[6] & Yuki Kimura[6]

[22]Department of Physics, The Catholic University of America, Washington, DC 20064, USA. [23]Division of Geological and Planetary Sciences, California Institute of Technology, Pasadena, CA 91125, USA. [24]Earth-Life Science Institute (ELSI), Tokyo Institute of Technology, Tokyo 152-8550, Japan. [25]Planetary Exploration Research Center, Chiba Institute of Technology, Narashino 275-0016, Japan. [26]Université Paris-Saclay, CNRS, Institut de Chimie Physique, Orsay 91405, France. [27]Department of Life, Environment and Material Science, Fukuoka Institute of Technology, Fukuoka 811-0295, Japan. [28]HORIBA Advanced Techno, Co., Ltd., Kisshoin, Minami-ku, Kyoto 601-8510, Japan. [29]HORIBA Techno Service Co., Ltd. Kisshoin, Minami-ku, Kyoto 601-8510, Japan. [30]HORIBA Techno ServiceCo., Ltd, Kyoto 601-8305, Japan. [31]Geological Survey of Japan (GSJ), National Institute of Advanced Industrial Science and Technology (AIST), 1-1-1 Higashi, Tsukuba, Ibaraki 305-8567, Japan. [32]LECO Japan Corp, Tokyo 105-0014, Japan.

