## [Peer Review File · Nature Communications]

Primordial aqueous alteration recorded in water-soluble organic molecules from the carbonaceous asteroid (162173) RyuguReviewer #1 (Remarks to the Author):

Review of "Primordial aqueous alterations recorded in water-soluble organic molecules from the carbonaceous asteroid (162173) Ryugu"

Let me start by stating that both scientific and lay audiences must be given the opportunity to read through the communication submitted by Takano et al, which deals with primordial aqueous alteration within the carbonaceous asteroid Ryugu, as recorded in water soluble molecules within the sample returned by JAXA's Hayabusa2 spacecraft. Without question, the Hayabusa2 mission rivals the Apollo sample return missions of the late 1960s early 1970s in terms of its contributions to our understanding of life's origin and the evolution of the Solar System. It is without equivocation, therefore, that I fully endorse publishing this important study in Nature Communications, once the changes/questions I have provided below have been addressed by the author team.

The authors have, in my opinion, constructed a plausible argument for how oxidized, water-soluble species such as dicarboxylic acids, tricarboxylic acids and hydroxycarboxylic acids are molecular records of the aqueous alteration that occurred within Ryugu. Moreover, many of these species are prebiotic molecules that have the potential to become incorporated into living systems, so their analysis in this paper can inform us on how life may have started on the Earth.

The authors interpretation of the data is chemically sound and the conclusions posited are entirely reasonable. Of particular note is their successful linkage of secondary mineral assemblages and aqueously altered vein formations to the suite of hydrophilic, oxidized organic species found in the hot water extracts.

I was also pleased to see that their focus was on dicarboxylic acids, tricarboxylic acids and hydroxy carboxylic acids, which have received much less attention than monocarboxylic acids, presumably because they are typically found in much lower concentrations than monocarboxylic acids (e.g. formic acid and acetic acid) in carbonaceous chondrites with appreciable levels of aqueous alteration.

In addition, members of the meteoritical who perform trace analysis studies will gain some insight and perhaps inspiration from the authors use of capillary electrophoresis to separate hydroxy, di and tricarboxylic acids. To the best of my knowledge, capillary electrophoresis, while widely used for the separation of amino acids, has been employed only sparingly, until now, by researchers for the determination and quantification of carboxylic acids in meteorites.

Overall, the text is clear, and the arguments presented are well constructed. The authors have done a laudable job of citing relevant, previous publications in their paper. There are no sections within the short communication that should be excised, in my opinion.

My perusal of the communication itself and the supplementary information uncovered a few grammatical issues that should be addressed before the paper is published. My suggested minor changes to the text/figure captions are itemized below, along with their locations within the two parts of the submission. Please note that my suggested edits are in bold.

Main Body

Page 3, line 81: Pristine samples from the near-Earth asteroid (162173) Ryugu returned to Earth ...

Page 3, line 87: What are the origins and characteristics of the light elements...

Page 4, line 103: between the organic and inorganic interface...

Page 4, line 111: contact between hot water and pristine Ryugu samples.

Page 4, line 127: and hydrophilic molecular groups in the hot water extracts...

Page 5, line 136: To which properties of lactic acid are the authors referring?

Page 5, line 160: This sentence is a bit confusing to me. What unique characteristics are exhibited by malonic acid with respect to its keto-enol tautomerization?

Page 8, line 237: will be important for describing nitrogen dynamics... (remove the an after be)

Page 8, line 241: When investigating the history of the carbonaceous asteroid (162173)...

Page 8, line 251: spacecraft has returned the carbonaceous asteroid...

Page 8, line 253: scientific opportunities to explore the history of organic chemical evolution (remove the word pristine).

Page 8, line 254: We hope that the Bennu sample will reveal...

Page 9, line 267: the project team performed an environmental evaluation...

Page 9, line 272: photographed (this study)...

Page 11, line 341: After discovering the yellow sticky deposit on the wall in the glass vial

containing the IOM fraction...

Page 11, line 342: we conducted an n-hexane extraction...

Page 18, line 555: Supplementary Information is available for this paper.

Page 22, line 610: The blank was composed of ultrapure water before...

Page 22, line 617: is shown in the same formulation.

Page 23, line 623: keto-enol tautomerism of malonic acid, ...

Page 28, line 674: Figure 8. Aqueous alteration of primordial hydrophilic...

Page 29, line 692: hydrothermally produced precipitates...

Supplementary Material

Page 32, line 772: to describe our...? Some words are missing here.

Page 32, line 781: glass ampoules. After the extraction, the contents were transferred from...

Page 33, line 796: please refer to the reference (Naraoka+2023)

Page 33, line 800: using the same procedure as in the present report.

Page 35, line 871: shows that the Ryugu samples...

Page 36, line 895: which could be caused by residual water on the parent body.

Page 38, line 907: Strategies for exploring the carbonaceous asteroid...

Page 38, line 915: and a comprehensive...

Page 41, line 947: color scale in the Venn diagram stands for...

Page 47, line 991: cyclic sulfur molecules in the IOM leachate.

Page 54, line 1035: stable isotopic compositions for the Ryugu IOM solid residues.

Although the paper is quite solid, there is one outstanding issue that the authors should address in order to provide a better link to the results published by Naraoka et al earlier this year (viz., *Science*, 379, February 24, 2023, pp. 1-10, eabn9033) and to further contextualize the findings of their important study. Here, I am referring to the lack of any discussion of the monocarboxylic acids in the hot water extracts that were analyzed by capillary electrophoresis (CE)-high resolution mass spectrometry (HRMS). In fact, the authors state in their introduction that they comprehensively evaluated highly diverse hydrophilic organic molecules using capillary electrophoresis coupled to high resolution mass spectrometry and yet they do not mention detecting any monocarboxylic acids. This is surprising in light of the fact that Naraoka et al in their *Science* paper (vide supra) found substantial quantities of both formic and acetic acid in the hot water extract of the Ryugu sample A0106. Naraoka et al stated quite correctly in their *Science* paper that light carboxylic acids (e.g. formic acid, acetic acid, propanoic acid) are among the most abundant organic compounds in organic-rich carbonaceous chondrites (and hence their parent bodies) that have undergone low-temperature, hydrothermal processing. It is worth pointing out here that there are papers in the literature in which capillary electrophoresis was successfully used to separate and quantify light monocarboxylic acids in aqueous extracts (see for instance: Separation and determination of some carboxylic acids by capillary electrophoresis Vladimir SLADKOV, Blandine FOUREST Institute of Nuclear Physics, CNRS/IN2P3, 91406 Orsay, France). Consequently, I suspect that the capillary electrophoresis work conducted by Takano et al also uncovered detectable quantities of light monocarboxylic acids, unless the method they used cannot detect light monocarboxylic acids. Of course, it could be that the reason the authors have not mentioned signals for formic and acetic acid, the two most abundant monocarboxylic acids found by Naraoka et al as their propyl derivatives using a GC-MS method, is because the peaks for these acids were swamped by the acetate buffer and thus were not clearly visible. If this is, indeed, their conclusion, they should mention this in the paper as informed readers might be confused by the fact that these light carboxylic acids were found in high concentrations using a GC-MS method on their derivatives while they were apparently not detected using capillary electrophoresis, which did, nonetheless, find peaks for hydroxy carboxylic acids, dicarboxylic acids and tricarboxylic acids that are almost certainly present in much lower abundances than the light monocarboxylic acids, as is typically the case for aqueously altered carbonaceous chondrites. If Takano et al look back at the capillary electrophoresis data and find they can determine the peak area for any formic acid (or perhaps propanoic acid) present, it would be interesting to compare their calculated concentration for formic acid to that found by Naraoka et al to see how closely the two analytical methods agree. (I am assuming that acetic acid cannot be quantified because the

buffer used for the electrophoresis contains acetate and acetic acid).

Thank you for the opportunity to review this important communication. The shortcomings of the work are small and easily fixed. I look forward to the authors responses to my comments/suggested changes.

Most sincerely,
Robert Hilts

Reviewer #2 (Remarks to the Author):

Thank you for inviting me to review this interesting manuscript. This manuscript provides significant data on the water soluble organic content of asteroid Ryugu samples, with new data showing the presence and abundances of hydroxyl and dicarboxylic acids. I have provided my comments and suggested edits on the attached PDF file so that it is clear where my comments refer to. The manuscript provides a discussion on the hydroxyl and dicarboxylic acid contents, and how these compare to the amino acids and IOM contents of asteroid Ryugu from recently published papers. There are a couple of areas in which the discussion can be strengthened. For example, discussion on previous amino acid analyses and how that enhance our interpretation on the aqueous alteration extent of Ryugu was present, but further discussion and comparison to the correlation to other water soluble molecules, in particular the hydroxyl and dicarboxylic acids presented in this study, should be added (i.e. the session "The systematics of hydrophilic molecules at two sampling locations on Ryugu" can be enhanced). There is a description on the elemental and hydrophilic molecule compositions between TD1 and TD2, however the discussion comparing and contrasting between the 2 sites are not fully developed. Overall, the data presented in the manuscript is of significant importance to the community of planetary science and cosmochemistry and thus should be published.

In their review of the first version of this manuscript, reviewer #2 added some comments to the manuscript file. These comments were forwarded to the authors, who replied as included in this Peer Review File.

Reviewer #3 (Remarks to the Author):

Dear Authors,

This study focus on the analysis of hot water extract from Ryugu returned samples in order to understand the effect of aqueous alteration on the molecular diversity of polar organic molecules extracted. The authors postulate that these polar molecules should have recorded traces of this aqueous alteration in Ryugu. Results of the analysis of polar molecules are presented in the result section and discussed with results from other papers in regard to the aqueous alteration history. Overall, the presented results are very dense, and under its current form the paper is quite confused (not only regarding the figure referencing). I found that the paper intend to present and discuss the results with readers who are perfectly aware of the results from other papers on Ruygu organic matter analysis. As a result, the paper is not for a broad audience, that will not see the interest and the main results of the paper. Notably, a lot of interesting results are in the supplementary part, and not at all discussed in the main text. figure S9 to S14 not cited in the main text and Fig S10 to S14 neither in the supplementary files. Right now the paper looks more like a draft, where the authors wanted to put all the unpublished figures from previous analysis (and previous papers) and try to find relation between them. I am not agree for the publication of the paper under this form. I suggest either the paper clearly states that it will be a huge discussion and wrap up comparison of all previous studies, including the new analysis of polar compounds to

search for the record of aqueous alteration on the organics, or the authors focus only on the polar compounds detected in this #7-1 extract and discuss the relevance of such compounds by comparison to others meteorite studies. Right now the abstract and the intro is only focus on the polar compounds, and the more we read the paper, the more we understand that it is not only that, and the discussion and all the figures (not enough discussed) lead the reader way more further but is a confuse way, and highlights are not well written.

Please see the pdf attached for more detailed comments on the paper.

Reviewer #3 Attachment on the following page

Primordial aqueous alterations recorded in water-soluble organic molecules from the carbonaceous asteroid (162173) Ryugu.

This study focus on the analysis of hot water extract from Ryugu returned samples in order to understand the effect of aqueous alteration on the molecular diversity of polar organic molecules extracted. The authors postulate that these polar molecules should have recorded traces of this aqueous alteration in Ryugu. Results of the analysis of polar molecules are presented in the result section and discussed with results from other papers in regard to the aqueous alteration history. Overall, the presented results are very dense, and under its current form the paper is quite confused (not only regarding the figure referencing). I found that the paper intend to present and discuss the results with readers who are perfectly aware of the results from other papers on Ruygu organic matter analysis. As a result, the paper is not for a broad audience, that will not see the interest and the main results of the paper. Notably, a lot of interesting results are in the supplementary part, and not at all discussed in the main text. figure S9 to S14 not cited in the main text and Fig S10 to S14 neither in the supplementary files. Right now the paper looks more to a draft, where the authors wanted to put all the unpublished figures from previous analysis (and previous papers) and try to find relation between them. I am not agree for the publication of the paper under this form. I suggest either the paper clearly states that it will be a huge discussion and wrap up comparison of all previous studies, including the new analysis of polar compounds to search for the record of aqueous alteration on the organics, or the authors focus only on the polar compounds detected in this #7-1 extracts and discuss the relevance of such compounds by comparison to others meteorite studies. Right now the abstract and the intro is only focus on the polar compounds, and the more we read the paper, the more we understand that it is not only that, and the discussion and all the figures (not enough discussed) lead the reader way more further but is a confuse way, and highlights are not well written. Overall, it is not suitable for a Nature.

Main issues in addition:

1/The identification and quantification of polar organic compounds is problematic, as the identification does not take into account the matrix effects that are important in such samples, particularly in terms of retention time. How is it possible to identify these molecules with exact retention times and masses, knowing that the retention times are not the same compared to the standard times due mainly to the matrix effects ?

A table is also missing, reporting the retention time and masses from standards and molecules identified in the samples.

2/ The idea of malonic acid/ acetic acid as a clue for the degree of aqueous alteration is interesting, however, acetic acid has not been reported as detected compounds, how is that possible ? Furthermore, linking the malonic acid/acetic acid ratio to conditions of aqueous alteration would appear to be flawed, since acetic acid also likely has other formation pathways independent of the tautomerisation of malonic acid. So unless you can show that this ratio is relevant for many samples, I have great doubts about the theory. Also in Figure 4, the two molecules in brackets are the same, the tautomerisation is between the keto and enol form. It should be written below malonic acid, that it is the keto-form.

3/As already said, this paper is not clear with a lot of misquoted figures and given the large amount of data, it is difficult to understand what was done in this study, from what was done in the previous studies. The authors should more emphasize their work.

4/ the introduction is again for specialists, and only focus on the Ryugu previous analysis, while it's been decades that chondrites coming from asteroids (like Ryugu) are analysed in the lab. From line 98 to 109 authors wrote a list of all the analysis that was done in Ryugu without objectives concerning their work.

5/ finally, after reading the manuscript, I think the title is not appropriate, since none of their data confirm the effect of aqueous alteration. Some for figure 8, which is great but not supported by the results.

Specific comments from line to line or in figures:

Main text:

Figure 1: the figure is very nice, but do contain a lot of information, which are not considered/discussed here. Is it useful ?

Lin 87: I don't think the question regarding Ryugu's role in the solar system history is relevant.

Figure 2: pyruvic acid is not mentioned in the legend.

Line 113: why the aqueous alteration should be only recorded by the polar organic molecules ? please rephrase.

Line 127: write the analytical technique from which you obtained the identification of hydroxyl acids. And figure 2 should be cited here.

Line 128: sample name (#7-1) the name of the sample appears from nowhere, lacking an explanation (we have to go in the SI, looking at Fig S2 !). once again, you need to appeal to a wide audience for a Nature paper.

Line 128: After giving the name of the hot water fraction, it would be a good idea to refer to figure S3 detailing all the extracts.

Line 129: Are the authentic standards those in figure S4? Also in the figure 3-A, please add the standard signal. It is stated in the text that identification is possible using the retention time and the exact mass. However, if we refer to figure S4 and compare the retention times of the standards with figure 3-A, it is impossible to identify the compounds in Ryugu extracts.

Line 129, Also, what is an authentic standard? can we have factice standards??

Figure 3-C, please specify in legend which molecules were used in the “ α -hydroxy- acids (Cn) in Murchison”.

Line 133: How was the quantification carried out? External calibration on standards is a very uncertain method due to the presence of salts and the famous matrix effect on the detected compounds.

Table S1: Please specify error bars, also specify in the legend abbreviations such as "n.d". Not all the compounds in figure 2 are shown, but some that are never mentioned are.

Line 139: pyruvic acid is not included in table S1

Line 147-148: Urea and glycoamine are not included in table S1.

Line 151: please specify where in the SI. In the paragraph "Spectroscopic FTIR references for the soluble components", these bands are not mentioned, nor are they mentioned in figure S9, although it would appear that they are mentioned in figure S13 (which is in fact never cited).

Figure 4-A: Does the x-axis "dicarboxylic acid" take into account all the compounds detailed on lines 157-158? Please specify in the legend.

Figure 4-B: The two-enol forms of malonic acid are in fact the same, since the molecule is symmetrical.

Line 168- Please refer to figure 4-B.

Line 172: please smooth this sentence, as other ways of forming acetic acid can take place in these objects and therefore modify the malonic/acetic ratio, as the formation pathway from malonic acid or acetic acid under aqueous alteration conditions is not the only one.

Figure 5: in the legend (line 635), Table 1 does not exist; the reference should be Table S1. Please clearly state what is new and your work here. Are all the plots in this figure really relevant to this study? IB is not explained.

Line 188: Where are the data for the figure 5-F about the urea molecules?

Line 190: ammonium ions has been proposed has the main salt component of comet 67P, please refer to this study also.

From line 197 to 211, the authors discussed the detection of amino acids in CM and CI, compared to Ryugu, as these class of meteorites where the same. This is not true, and it is quite known now that parent bodies of CM and CI has not accreted the same pool of material from the nebula, including the organics. And of course, they has not the same aqueous/thermal history. So it is questionable to compare quantities and identification of amino acids from group to group.

Line 203: the reference Burton is not correctly cited.

Line 237: missing words

Figure 6-A: Are all the molecules detected and reported in table S1 ?

Line 244: Misquotation of figure.

Method section:

Line 288: Why is Figure 4 cited here ?

Line 275: In this paragraph, add the origin of the different standards and their purity.

Line 304: Misquotation of figure, should be Figure 1-G.

Line 350: Please specify what is referred to in the supplementary information, as figures S-10 to S-14 are never mentioned.

Supporting info

Line 771: in Figure S1 only the sample C0107 is presented.

Line 772 to 773, is there a missing word ?

Figure S3: in the legend, please add the description for C.

Figure S5: line 950 in the legend, misquotation of figure it should be Figure S9 rather than Figure 9.

Line 812: the name “remaining nitrogen” is misleading. The amount of nitrogen was summed after each successive extraction and taken as the total amount in the bulk (initial bulk as 100%). So this is not the remaining nitrogen but the nitrogen extracted by each solvent extraction. The remaining nitrogen is the one that remains in the non-soluble part (IOM) .

Line 836: The extract #7-1 is called hot H₂O extract in the text but H₂O-HCl extracts in the legend of Figure S7.

Figure S7, please add name of the axes for all “lines of plots”.

Figure S9, name of the axes ?

Line 967: For the formic acid extract, please add the blank in the Figure S8.

Figure S12-B: this figure is not cited in the paper or the SI, nor discussed anywhere and, contains a huge mistake. The Kovats relation is linear for $\log(t_r')=f(C\text{-number})$, the current representation is wrong. Please correct.

Replies to Reviewer's comments on the manuscript # NCOMMS-23-43936

We appreciate the constructive review comments from the associate editor and three reviewers on our manuscript (# NCOMMS-23-43936) entitled "Primordial aqueous alterations recorded in water-soluble organic molecules from the carbonaceous asteroid (162173) Ryugu". We carefully read **the reviewing comments in blue** (#1: Dr. Robert Hilts, #2 & #3: anonymous) and modified all of the concerns in the original version of the manuscript. The revision we made based on the reviewer's comments are improved **in red** on the main manuscript and supporting information. As noted in the revised manuscript, we acknowledge the constructive feedback from the three reviewers.

Here, we note that we already declared the option of a "transparent peer review system" on the initial paper submission in order to make public the discussions and feedback from the peer review process (*Nature Communications*, doi: 10.1038/ncomms10277). According to that publisher's rule, therefore, all comments from the reviewers will also be made public on the journal site.

Reviewer #1 (Remarks to the Author):

Let me start by stating that both scientific and lay audiences must be given the opportunity to read through the communication submitted by Takano et al, which deals with primordial aqueous alteration within the carbonaceous asteroid Ryugu, as recorded in water soluble molecules within the sample returned by JAXA's Hayabusa2 spacecraft. Without question, the Hayabusa2 mission rivals the Apollo sample return missions of the late 1960s early 1970s in terms of its contributions to our understanding of life's origin and the evolution of the Solar System. It is without equivocation, therefore, that I fully endorse publishing this important study in *Nature Communications*, once the changes/questions I have provided below have been addressed by the author team.

The authors have, in my opinion, constructed a plausible argument for how oxidized, water-soluble species such as dicarboxylic acids, tricarboxylic acids and hydroxycarboxylic acids are molecular records of the aqueous alteration that occurred within Ryugu. Moreover, many of these species are prebiotic molecules that have the potential to become incorporated into living systems, so their analysis in this paper can inform us on how life may have started on the Earth.

The authors interpretation of the data is chemically sound and the conclusions posited are entirely reasonable. Of particular note is their successful linkage of secondary mineral assemblages and aqueously altered vein formations to the suite of hydrophilic, oxidized organic species found in the hot water extracts. I was also pleased to see that their focus was on dicarboxylic acids, tricarboxylic acids and hydroxy carboxylic acids, which have received much less attention than monocarboxylic acids, presumably because they are typically found in much

lower concentrations than monocarboxylic acids (e.g. formic acid and acetic acid) in carbonaceous chondrites with appreciable levels of aqueous alteration.

In addition, members of the meteoritical who perform trace analysis studies will gain some insight and perhaps inspiration from the authors use of capillary electrophoresis to separate hydroxy, di and tricarboxylic acids. To the best of my knowledge, capillary electrophoresis, while widely used for the separation of amino acids, has been employed only sparingly, until now, by researchers for the determination and quantification of carboxylic acids in meteorites. Overall, the text is clear, and the arguments presented are well constructed. The authors have done a laudable job of citing relevant, previous publications in their paper. There are no sections within the short communication that should be excised, in my opinion.

[Replies] The authors are greatly encouraged by the above comments. We would like to express our deep gratitude for the careful reading of our manuscript. In this revised manuscript, several references were updated to the latest information.

Schmitt-Kopplin, P. et al. Soluble organic matter Molecular atlas of Ryugu reveals cold hydrothermalism on C-type asteroid parent body. *Nature Commun.* **14**, Article number: 6525 (2023).

Yada, T. et al. A curation for uncontaminated Hayabusa2-returned samples in the extraterrestrial curation center of JAXA: from the beginning to present day. *Earth Planets Space* **75**, Article number: 170 (2023).

Yoshimura, T. et al. Chemical evolution of primordial salts and organic sulfur molecules in the asteroid (162173) Ryugu. *Nature Commun.* **14**, Article number: 5284 (2023).

Zeichner, S. S. et al. Polycyclic aromatic hydrocarbons in samples of Ryugu formed in the interstellar medium. *Science* **382**, 1411-1416.

My perusal of the communication itself and the supplementary information uncovered a few grammatical issues that should be addressed before the paper is published. My suggested minor changes to the text/figure captions are itemized below, along with their locations within the two parts of the submission. Please note that my suggested edits are in bold.

[Replies] The authors deeply appreciate the careful suggestions. We made minor corrections and grammatical refinements according to the reviewing comments.

Main Body

Page 3, line 81: Pristine samples from **the** near-Earth asteroid (162173) Ryugu returned to Earth..

Page 3, line 87: What are the origins and characteristics of **the** light elements...

[Replies] Minor revisions for page 3 were made in accordance with review comments.

Page 4, line 103: between **the** organic and inorganic interface...

Page 4, line 111: contact between hot water and **pristine** Ryugu samples.

Page 4, line 127: and hydrophilic molecular groups **in** the hot water extracts...

[Replies] Minor revisions for page 4 were made in accordance with review comments.

Page 5, line 136: **To which properties of lactic acid are the authors referring?**

[Replies] We revised in the line 136 as “ The concentration of lactic acid (C₃),..”.

Page 5, line 160: **This sentence is a bit confusing to me. What unique characteristics are exhibited by malonic acid with respect to its keto-enol tautomerization?**

[Replies] The part was corrected to "water sensitive properties", which is a plainer phrase.

Page 8, line 237: will be important for describing nitrogen dynamics... (remove the **an** after be)

[Replies] Removed as suggested.

Page 8, line 241: When **investigating** the history of the carbonaceous asteroid (162173)...

Page 8, line 251: spacecraft **has returned** the carbonaceous asteroid...

Page 8, line 253: scientific opportunities to explore the history of organic chemical evolution (remove the word **pristine**).

Page 8, line 254: We hope that the **Bennu** sample will reveal...

[Replies] All minor revisions for page 8 were made in accordance with review comments.
The reference of Foustoukos+2024 was mentioned as important perspectives.

[Ref.]

Foustoukos, D. et al. Bulk H, C, and N in Samples Returned from Asteroid Bennu: Comparison with Carbonaceous Chondrites, *Lunar and Planetary Science Conference (LPSC)*, #1190 (2024).

Page 9, line 267: the project team performed **an** environmental evaluation...

Page 9, line 272: **photographed** (this study)...

[Replies] All minor revisions for page 9 were made in accordance with review comments.

Page 11, line 341: After **discovering** the yellow sticky deposit **on the wall** in the glass vial **containing** the IOM fraction...

Page 11, line 342: we conducted **an** n-hexane extraction...

[Replies] All minor revisions for page 11 were made in accordance with review comments.

Page 18, line 555: Supplementary Information **is** available for this paper.

[Replies] Minor revision for page 18 were made in accordance with review comments.

Page 22, line 610: The blank was **composed of** ultrapure water before...

Page 22, line 617: **is** shown in the same formulation.

[Replies] All minor revisions for page 22 were made in accordance with review comments.

Page 23, line 623: keto-enol tautomerism of malonic acid, ...

[Replies] Minor revision for page 23 were made in accordance with review comments.

Page 28, line 674: Figure 8. Aqueous alteration **of** primordial hydrophilic...

Page 29, line 692: hydrothermally **produced** precipitates...

[Replies] All minor revisions for page 28&29 were made in accordance with review comments.

Supplementary Material

Page 32, line 772: to describe our...? **Some words are missing here.**

[Replies] We revised as “We present a concept based on the dimensions of chemical resolution and chemical variation (Figure S2).”

Page 32, line 781: glass **ampoules**. After the extraction, the **contents were** transferred from...

[Replies] Minor revisions for page 32 were made in accordance with review comments.

Page 33, line 796: please refer to the **reference** (Naraoka+2023)

Page 33, line 800: using the same procedure **as** in the present report.

[Replies] All minor revisions for page 33 were made in accordance with review comments.

Page 35, line 871: **shows that the Ryugu samples...**

Page 36, line 895: which could be caused by **residual water** on the parent body.

[Replies] All minor revisions for page 35&36 were made in accordance with review comments.

Page 38, line 907: Strategies for exploring **the carbonaceous asteroid...**

Page 38, line 915: and **a comprehensive...**

[Replies] All minor revisions for page 38 were made in accordance with review comments.

Page 41, line 947: color scale in **the Venn diagram** stands for...

Page 47, line 991: cyclic sulfur molecules in **the IOM leachate**.

Page 54, line 1035: stable isotopic compositions **for the Ryugu IOM solid residues**.

[Replies] All minor revisions for page 41, 47, 54 were made in accordance with review comments.

Although the paper is quite solid, there is one outstanding issue that the authors should address in order to provide a better link to the results published by Naraoka et al earlier this year (viz., Science, 379, February 24, 2023, pp. 1-10, eabn9033) and to further contextualize the findings of their important study. Here, I am referring to the lack of any discussion of the monocarboxylic acids in the hot water extracts that were analyzed by capillary electrophoresis (CE)-high resolution mass spectrometry (HRMS). In fact, the authors state in their introduction that they comprehensively evaluated highly diverse hydrophilic organic molecules using capillary electrophoresis coupled to high resolution mass spectrometry and yet they do not mention detecting any monocarboxylic acids. This is surprising in light of the fact that Naraoka et al in their Science paper (vide supra) found substantial quantities of both formic and acetic acid in the hot water extract of the Ryugu sample A0106. Naraoka et al stated quite correctly in their Science paper that light carboxylic acids (e.g. formic acid, acetic acid, propanoic acid) are among the most abundant organic compounds in organic-rich carbonaceous chondrites (and hence their parent bodies) that have undergone low-temperature, hydrothermal processing. It is worth pointing out here that there are papers in the literature in which capillary electrophoresis was successfully used to separate and quantify light monocarboxylic acids in aqueous extracts (see for instance: Separation and determination of some carboxylic acids by capillary electrophoresis Vladimir SLADKOV, Blandine FOUREST Institute of Nuclear Physics, CNRS/IN2P3, 91406 Orsay,

France). Consequently, I suspect that the capillary electrophoresis work conducted by Takano et al also uncovered detectable quantities of light monocarboxylic acids, unless the method they used cannot detect light monocarboxylic acids. Of course, it could be that the reason the authors have not mentioned signals for formic and acetic acid, the two most abundant monocarboxylic acids found by Naraoka et al as their propyl derivatives using a GC-MS method, is because the peaks for these acids were swamped by the acetate buffer and thus were not clearly visible. If this is, indeed, their conclusion, they should mention this in the paper as informed readers might be confused by the fact that these light carboxylic acids were found in high concentrations using a GC-MS method on their derivatives while they were apparently not detected using capillary electrophoresis, which did, nonetheless, find peaks for hydroxy carboxylic acids, dicarboxylic acids and tricarboxylic acids that are almost certainly present in much lower abundances than the light monocarboxylic acids, as is typically the case for aqueously altered carbonaceous chondrites. If Takano et al look back at the capillary electrophoresis data and find they can determine the peak area for any formic acid (or perhaps propanoic acid) present, it would be interesting to compare their calculated concentration for formic acid to that found by Naraoka et al to see how closely the two analytical methods agree. (I am assuming that acetic acid cannot be quantified because the buffer used for the electrophoresis contains acetate and acetic acid).

[Replies] I appreciate the valuable comments. Monocarboxylic acids such as formic acid and acetic acid have been shown in previous reports (Naraoka et al., 2023). Here, we have additionally reanalyzed a series of monocarboxylic acid molecules that were previously unidentified. Based on the results of our reanalysis, the molecular groups of monocarboxylic acids were of diverse chemical compositions, including aliphatic, aromatic, unsaturated, and keto acids, as shown in Figure 2. In this context, Table S1 was updated to show the identification of molecules and the assessment of separation conditions by CE-HRMS in Figure S4.

Since new data have been added to the present report, the title of the paper has been slightly revised as follows. “Primordial aqueous alterations recorded in water-soluble organic molecules of hydroxy acids and carboxylic acids from the carbonaceous asteroid (162173) Ryugu.”

As the reviewer pointed out, formic acid and acetic acid are difficult to quantify due to the analytical conditions of the capillary electrophoresis. Please see the method section of “Capillary Electrophoresis.: Capillary electrophoresis was carried out using a CE System G1600AX (Agilent Technologies Inc., Santa Clara, CA), and an LC-30 AD pump (Shimadzu Corporation, Kyoto, Japan) was used to create the sheath flow. The sheath-flow rate was set at 10 μ L min. Metabolites were separated using a fused silica capillary \times (50 μ i.d. 80 cm total length; Polymicro Technologies, Inc., Phoenix, AZ) with 50 mM ammonium acetate (pH 8.5) for anion analysis and 1 M formic acid for cation analysis. The applied voltage for CE was set at 30 kV. The systems were controlled by ChemStation

software version B.04.03 (Agilent Technologies Inc.) Mass Spectrometry.”, and the importance of formic acid parameter on CE condition (Salzer+2023).

[Ref.]

Salzer, L. et al. Capillary electrophoresis-mass spectrometry as a tool for *Caenorhabditis elegans* metabolomics research. *Metabolomics* **19**, Article number: 61 (2023).

Thank you for the opportunity to review this important communication. The shortcomings of the work are small and easily fixed. I look forward to the authors responses to my comments/suggested changes.

Most sincerely,
Robert Hilts

[Replies] The authors have addressed all comments point-by-point and have advanced this response letter. The revised manuscript has also been updated as appropriate, along with the requested raw data profiles for aliphatic, aromatic, unsaturated, and keto-form monocarboxylic acids. We believe that these primary data represent a valuable opportunity to report new organic astrochemical findings from the carbonaceous asteroid Ryugu.

We also expect that this knowledge will be an important baseline for the science of the carbonaceous asteroid Bennu and the OSIRIS-REx (Origins, Spectral Interpretation, Resource Identification, and Security – Regolith Explorer) sample return mission.

Reviewer #2 (Remarks to the Author):

Thank you for inviting me to review this interesting manuscript. This manuscript provides significant data on the water soluble organic content of asteroid Ryugu samples, with new data showing the presence and abundances of hydroxyl and dicarboxylic acids. I have provided my comments and suggested edits on the attached PDF file so that it is clear where my comments refer to. The manuscript provides a discussion on the hydroxyl and dicarboxylic acid contents, and how these compare to the amino acids and IOM contents of asteroid Ryugu from recently published papers. There are a couple of areas in which the discussion can be strengthened. For example, discussion on previous amino acid analyses and how that enhance our interpretation on the aqueous alteration extent of Ryugu was present, but further discussion and comparison to the correlation to other water soluble molecules, in particular the hydroxyl and dicarboxylic acids presented in this study, should be added (i.e. the session "The systematics of hydrophilic molecules at two sampling locations on Ryugu" can be enhanced). There is a description on the elemental and hydrophilic molecule compositions between TD1 and TD2, however the discussion comparing and contrasting between the 2 sites are not fully developed. Overall, the data presented in the manuscript is of significant importance to the community of planetary science and cosmochemistry and thus should be published.

[Replies] The authors are greatly encouraged by the above comments. We would like to express our sincere acknowledgement for the careful reading of our manuscript. We made appropriate corrections based on the reviewing comments, especially for the correlations with hydroxy acids and organic acids described in this study. We have revised and added a discussion of comparative considerations between the two locations as shown below.

L71-72: Please provide some specificity here, e.g. relevant to prebiotic molecular evolution such as the formation of XXXX products.

L99-100: spellings of these elements should appear in line 88 when they first appear

[Replies] Revised as "...relevant to prebiotic molecular evolution such as the primordial TCA (tricarboxylic acid) cycle."

We agree and added those indications in the line 88-95.

L108: SOM

[Replies] Revised as suggested.

L108: ~20000 = not clear what this number indicates, I assume the number of molecules found in Ryugu, but does this include isomers?

[Replies] For a more accurate description, we revised as following, “Based on Fourier transform-ion cyclotron resonance mass spectrometry (FT-ICR/MS) analysis, the SOM from Ryugu samples contained highly diverse organic molecules (~20,000 species) in the surface regolith sample (Naraoka+2023; Schmitt-Kopplin+2023).” We have also updated the reference information of Schmitt-Kopplin+2023.

[Ref.]

Schmitt-Kopplin, P. et al. Soluble organic matter Molecular atlas of Ryugu reveals cold hydrothermalism on C-type asteroid parent body. *Nature Commun.* **14**, Article number: 6525 (2023).

L110-111: the expression can be rewritten as something like "we determine the molecular diversity of polar organic molecules extracted from samples, and we use this information to interpret the aqueous alteration process asteroid Ryugu has experienced. "

[Replies] We agreed the point, and revised as suggested.

L113: it is not very appropriate to use "pristine" to describe "chemical evolution"

[Replies] Deleted the term “pristine” as suggested.

L117: not very clear what is indicated here, please elaborate.

[Replies] We have inserted a sentence of explanation in that line and revised it as follows, “Naraoka et al. (2023) reported organic molecular diversity from initial bulk (IB) to insoluble organic matter (IOM) in the sequential extraction process using hydrophilic to hydrophobic solvents. Here, we also report the unique color characteristics of the sequentially extracted fractions with the systematic variation in ¹³C- and ¹⁵N-isotopic profiles”

L129: please indicate what standards you used (this information should be in the main text)

[Replies] Based on comments from reviewers #1 and #3, revisions were made in Figure 3 and Figure S4 regarding the relationship between CE-HRMS electropherograms and

detection times, and identification procedures from reference standards as following, “Figure S4. Separation on CE-HRMS for Ryugu, carbonaceous reference standards (Murchison & Murray), and carboxylic acid standard. (A) High-resolution mass electropherograms for Ryugu (A0106 & C0107), Murchison, Murray and propionic acid standard. (B) Verification of migration time on high-resolution mass electropherograms for carboxylic acids (mono-, di-, carboxylic acids of alkyl-straight chain molecules < C10), between Murchison reference and Ryugu sample (A0106&C0107) on the theoretical 1:1 line. Ryugu A0106 and C0107 are indicated by blue and light blue legends, respectively.” The standard mixture including the working reagents for migration-time alignment (e.g., AM1, AM2, AM3, AM4, and AM5), and an internal standard for anion analysis (ISA) were prepared from an HMT metabolomics kit (Human Metabolome Technologies Inc., Tsuruoka, Japan) (e.g., Sasaki+2019; Kami+2013; Sugimoto+2010). Due to the word limit of the main script, reference articles were also used as follows, “We used the representative carbonaceous meteorite of Murchison (Naraoka+2023; Oba+2023b; Koga+2024), as a reference standard to confirm our qualitative evaluation of the sample matrix effects. We also revised as “with reference standards (Murchison meteorite; Method).” Along with the previous report, information on the standard samples is provided.

[Ref.]

Kami, K. et al. Metabolomic profiling of lung and prostate tumor tissues by capillary electrophoresis time-of-flight mass spectrometry. *Metabolomics* **9**, 444–453 (2013).

Oba, Y. et al. Uracil in the carbonaceous asteroid (162173) Ryugu. *Nature Commun.* **14**, Article number: 1292 (2023b).

Koga, T. et al. Abundant extraterrestrial purine nucleobases in the Murchison meteorite: Implications for a unified mechanism for purine synthesis in carbonaceous chondrite parent bodies. *Geochim. Cosmochim. Acta*, **365**, 253–265 (2024).

L142-143: do they have any other roles in the formation of other intermediate molecules, those that are smaller than lipids which are large polymers?

[Replies] At this time, the results of the primary chemical description are presented. As we have stated in the conclusion also, further astrochemical surveys will be on comparative considerations with pristine samples from the asteroid Bennu collected at OSIRIS-REx mission. Also, additional science proposals after the initial analysis will be advanced by the international announcement of opportunity (AO) in the Astromaterials Science Research Group (ASRG) of ISAS/JAXA.

L147: Could you please elaborate further here whether these amino- and imino-groups you have identified in the Ryugu samples are present as free molecules, or simply as amines for example, or as a functional group within larger polymers or within the structure of large IOMs?

[Replies] The sentence was revised by adding "in the hot water extracts" at the end. As Koga et al. (2024) have shown in validation experiments with Murchison meteorite, then, components in the hot water extracts are presumably the chemical form that is easily extractable.

[Ref.]

Koga, T. et al. (2024) Abundant extraterrestrial purine nucleobases in the Murchison meteorite: Implications for a unified mechanism for purine synthesis in carbonaceous chondrite parent bodies. *Geochim. Cosmochim. Acta*, **365**, 253–265 (2024).

L160-161: could you elaborate a little further here please? what kind of unique characteristics were observed, how that specifically compared with your observation in this study, and how these characteristics reflect details of the tautomerization process?

L164: it would be useful to give several examples of the enol-keto tautomer pairs (in addition to malonic acid), and their relative proportion (a numerical value describing their relative abundance), based on your observation.

[Replies] This is very important point on this report. Figure 4-B was revised to facilitate understanding of the reaction summary. Also, the following discussion was added to the lines as, "Enol malonic acid is presumed to decompose faster than other dicarboxylic acids because it produces a thermodynamically unstable carbon-carbon double bond (i.e., HO-C=CH-, vinyl alcohol group: Chiang+1987; Cederstav & Novak, 1994; Kleimeier & Kaiser, 2021) during the aqueous alteration as follows,

Hence, the formation of two vinyl alcohol groups on the intra-molecular malonic acid is probably more reactive (chemically unstable) than other dicarboxylic acids (Figure 4-A).

[Ref.]

Cederstav, A.K. & Novak, B.M. Investigations into the chemistry of thermodynamically unstable species. The direct polymerization of vinyl alcohol, the enolic tautomer of acetaldehyde. *J. Am. Chem. Soc.* **116**, 4073–4074 (1994).

Chiang, Y. et al. Vinyl alcohol. Generation and decay kinetics in aqueous solution and determination of the tautomerization equilibrium constant and acid dissociation constants of the aldehyde and enol forms. *J. Am. Chem. Soc.* **109**, 4000–4009 (1987).
Kleimeier, N.F. & Kaiser, R.I. Interstellar Enolization: Acetaldehyde (CH₃CHO) and Vinyl Alcohol (H₂CCH (OH)) as a Case Study. *ChemPhysChem* **22**, 1229–1236 (2021).

L167: very interesting indeed - could you elaborate on the physical conditions of such decarboxylation process, would that be within the aqueous alteration regime at similar PT conditions? My understanding is that decarboxylation might involve higher temperatures to breakdown the molecules. But if Ryugu has not been heated, this process could not have prevailed?

[Replies]

We answered this comment in the previous question with some important references. Regards to temperature factor, it is estimated that the maximum temperature experienced by Ryugu is probably less than 100°C (Yokoyama+2023; Okada+2020) as we discussed in the main script. The evolutionary history of cold aqueous alteration in the parent body is summarized in Schmitt-Kopplin+2023.

L175-183: The relationship between total CNHOS vs TD1/TD2 is not further discussed in subsequent paragraphs, this paragraph should be further expanded to include a discussion to your findings, or else you can remove this paragraph.

[Replies] We would like to re-confirm that no one has reported such a summary of Hayabusa2 initial analysis, including the relationship of SOM and IOM profiles for TD1 and TD2. The original discussion of total CNHOS is discussed in the revised Figure 7 and Figure S5. Based on comments from reviewer #1 and suggestions for additional analysis, additional properties and new data (especially linear, branched-chain, unsaturated, and aromatic monocarboxylic acid series) for TD1 and TD2 organic acids were included (Figure 3 and Figure S4). We also confirm that subsurface material sampling is an important scientific opportunity and anticipate progress in the near future, based on the results of Nishiizumi+2022 and Ryugu AO sample analysis (Koga et al., in preparation).

[Ref.]

Nishiizumi, K. et al. Exposure conditions of samples collected on Ryugu's two touchdown sites determined by cosmogenic nuclides ¹⁰Be and ²⁶Al. *Lunar and Planetary Science*

Conference (LPSC), #1777 (2022).

L187-189: I totally agree with the authors

[Replies] We appreciate it.

L209-211: This paragraph introduces the fantastic figure 5. However, there seems to be a over description of the amino acid composition rather than the hydroxyl, when discussing the overall hydrophilic molecules abundances in Ryugu. Could you draw any linkage between the separate interpretations from hydroxyl and amino acid analyses?

[Replies] As this is the first analysis of a carbonaceous asteroid sample for any scientist, the authors have evaluated it quite carefully and politely. A statistical evaluation of hydroxyl and amino acid analysis was performed to determine the relevance, and Figures 6-A and B are shown. In fact, reviewer #1 gave us a highly positive review of the well-organized discussion points in this report. We believe it is important to verify the intercomparison in the analysis of asteroid Bennu samples in the near future.

L217: Is this IB different from previously discussed IB (lines 118 and 180)? Because from here I notice the IB is IB of OM. However, in Lines 118 and 180, IB definition wasn't very clear, and from the current way of description in lines 118 and 180 it seems you are discussing the initial bulk of the sample (i.e. Ryugu sample bulk including silicates)

[Replies] We appreciate the point of view. To avoid misunderstandings, we have added explanations of Σ SOM and Σ IOM as follows, “The Σ SOM represents the sum of the components extracted in each process of sequential extraction whereas Σ IOM represents the sum of the insoluble organic fractions as detailed in the previous literature (Naraoka+2023; Yabuta+2023). Hence, we define IB to mean pristine bulk, which includes all inorganic matrices such as silicates and carbonates, and IOM to be the fraction that does not contain silicates.”

L218: It is important to introduce clearly from line 118 whether you indeed refer to different IBs (silicate including or silicate excluded), or IB is IB of OM.

[Replies] We also appreciate the point. We have additionally included the following sentence, “Here, we define IB as whole rock bulk, which includes all inorganic matrices such as silicates and carbonates, and IOM to be the fraction that does not contain silicates

(Sephton+2003).” To avoid misunderstandings, the graph in Figure 7-B has been revised for clarity on the terminology of IB. The following sentence was added for sulfur as, “In contrast, it is interesting to note that the sulfur isotopic composition ($\delta^{34}\text{S}$) converged to VCDT scale ($\sim 0\text{‰}$) both before and after solvent extraction.”

L225-227: Although the depletion of heavy isotopes in IOM has been previously discussed in the literature, it is good to also discuss this here. What do the authors think about the disparities between the isotopic composition of SOM and IOM? Does it indicate formation sequence? Their genetic relationship (or not)? and perhaps more importantly for the topic of this paper - what's the implication on aqueous processing?

[Replies] We believe this is a very important discussion and perspective on this point. For nitrogen, a two-component mixture model was presented in a previous report of (Okazaki+2023). In addition, Hashizume and co-workers, which is independent of this study, has shown valuable data that complement each other (Hashizume+2022). We expect that the suggested point will be summarized in the near future, as it will be an important science object in the analysis of the asteroid Bennu.

[Ref.]

Hashizume, K. et al. Nitrogen isotopes in Ryugu return samples revealed by the stepwise combustion analysis in comparison with CI falls and Antarctic finds. #Abstract S32-04. *Hayabusa symposium* (2022).

Fig. 2, Fig.4-B: Removing the blue background color (i.e. presenting only white background) would enhance the readability of the figure content.

[Replies] Revised as suggested.

Fig. 6: what do the elliptic outline indicate? (or maybe I have missed this information elsewhere?)

[Replies] We added as follows, “The standard deviation (1σ) is indicated by the error arc.”

Reviewer #3 (Remarks to the Author):

Dear Authors,

This study focus on the analysis of hot water extract from Ryugu returned samples in order to understand the effect of aqueous alteration on the molecular diversity of polar organic molecules extracted. The authors postulate that these polar molecules should have recorded traces of this aqueous alteration in Ryugu. Results of the analysis of polar molecules are presented in the result section and discussed with results from other papers in regard to the aqueous alteration history. Overall, the presented results are very dense, and under its current form the paper is quite confused (not only regarding the figure referencing). I found that the paper intend to present and discuss the results with readers who are perfectly aware of the results from other papers on Ruygu organic matter analysis. As a result, the paper is not for a broad audience, that will not see the interest and the main results of the paper. Notably, a lot of interesting results are in the supplementary part, and not at all discussed in the main text. figure S9 to S14 not cited in the main text and Fig S10 to S14 neither in the supplementary files. Right now the paper looks more like a draft, where the authors wanted to put all the unpublished figures from previous analysis (and previous papers) and try to find relation between them. I am not agree for the publication of the paper under this form. I suggest either the paper clearly states that it will be a huge discussion and wrap up comparison of all previous studies, including the new analysis of polar compounds to search for the record of aqueous alteration on the organics, or the authors focus only on the polar compounds detected in this #7-1 extract and discuss the relevance of such compounds by comparison to others meteorite studies. Right now the abstract and the intro is only focus on the polar compounds, and the more we read the paper, the more we understand that it is not only that, and the discussion and all the figures (not enough discussed) lead the reader way more further but is a confuse way, and highlights are not well written.

[Replies] The authors are deeply appreciative of the careful reading throughout. Then, we would like to state that the comments raised by the reviewer #3 have been largely addressed in the responses and revisions to the reviewer #1 (Robert Hilts) and the reviewer #2 (anonymous). Figures S9 through S14 were already linked to the main text as “Supplementary Information”, but we followed the comments and added specific links in the revised manuscript. We revised the argument around the polar molecule group (#7-1) with the scientific justification. The overall reorganization is as follows.

1/The identification and quantification of polar organic compounds is problematic, as the identification does not take into account the matrix effects that are important in such samples, particularly in terms of retention time. How is it possible to identify these molecules with exact

retention times and masses, knowing that the retention times are not the same compared to the standard times due mainly to the matrix effects ?

[Replies] We appreciate the comment. To address the matrix effect issues on the LC- and CE-analytical procedure, we used the most representative carbonaceous meteorite of Murchison (Naraoka+2023; Oba+2023b; Koga+2024) as a reference standard to ensure our qualitative evaluation. Based on the comments, capillary-electropherogram showing the separation and time of the reference standard with a diagram systematically showing the molecular series identified in the sample are shown in Figure S4. The representative relationship between Murchison reference and Ryugu's (A0106, C0107) migration times for monocarboxylic acids and dicarboxylic acids with linear structures ($< C_{10}$) is shown in the diagram. Landmark signals such as structural isomers (e.g., *o*-, *m*-, *p*-; α -, β -) and system peaks can be used effectively to support molecular identification (e.g., Figure 3-A). It is also important to validate the correlation of elution times for a series of compounds (e.g., Figure S4).

The standard mixture including the working reagents for migration-time alignment (e.g., AM1, AM2, AM3, AM4, and AM5), and an internal standard (iSTD) for anion analysis were prepared concurrently from the HMT metabolomics kit (Human Metabolome Technologies Inc., Tsuruoka, Japan) (Sasaki+2019). Also, please note that the #7-1 fraction was not treated with concentrated hydrochloric acid (HCl), which can cause significant matrix effects as the reviewer are concerned (e.g., Koga+2024; Oba+2016).

[Ref.]

Koga, T. et al. Abundant extraterrestrial purine nucleobases in the Murchison meteorite: Implications for a unified mechanism for purine synthesis in carbonaceous chondrite parent bodies. *Geochim. Cosmochim. Acta*, **365**, 253–265 (2024).

Oba, Y. et al. Deuterium fractionation on the formation of amino acids by photolysis of interstellar ice analogues containing deuterated methanol. *Astrophys. J. Lett*, **827**, L18 (2016).

2/The idea of malonic acid/ acetic acid as a clue for the degree of aqueous alteration is interesting, however, acetic acid has not been reported as detected compounds, how is that possible ? Furthermore, linking the malonic acid/acetic acid ratio to conditions of aqueous alteration would appear to be flawed, since acetic acid also likely has other formation pathways independent of the tautomerisation of malonic acid. So unless you can show that this ratio is relevant for many samples, I have great doubts about this theory. Also in Figure 4, the two molecules in brackets are the same, the tautomerisation is between the keto and enol form. It should be written below

malonic acid, that it is the keto-form.

[Replies] As we responded in our feedback to reviewer #1, we reported the detection of acetic acid from Ryugu samples (A0106 & C0107, #7-1) (Naraoka+2023). In the revised manuscript, the decrease in the relative concentration of malonic acid was mentioned and the ratio to acetic acid was deleted. An important point in Figure 4, the two identical molecules show a high frequency factor of equilibrium to the enol form. The keto form is written under malonic acid.

3/As already said, this paper is not clear with a lot of misquoted figures and given the large amount of data, it is difficult to understand what was done in this study, from what was done in the previous studies. The authors should more emphasize their work.

[Replies] In accordance with the comments, all links to figure numbers have been corrected. As we responded to the comments of reviewers #1 and #2, the authors have revised the justification for this report with updated references.

4/ the introduction is again for specialists, and only focus on the Ryugu previous analysis, while it has been decades that chondrites coming from asteroids (like Ryugu) are analysed in the lab. From line 98 to 109 authors wrote a list of all the analysis that was done in Ryugu without objectives concerning their work.

[Replies] Based on the previous report (e.g., Gounelle & Zolensky, 2014), we would like to emphasize again “the sample value of freshness” without any weathering on the Earth. The other reviewers (#1 & #2) are well aware of the scientific value of "pristine" samples that can only be obtained from asteroid sample returns, as described in previous reports (Tachibana+2022; Yada+2022; Okazaki+2022). In the revised manuscript (L98-109), we stated that the objective of the present study was to analyze in detail the properties and characteristics of soluble organic matter in the “freshest” carbonaceous asteroidal samples.

[Ref.]

Gounelle, M., Zolensky, M.E. (2014) The Orgueil meteorite: 150 years of history. *Meteoritics & Planetary Science* 49, 1769-1794.

5/ finally, after reading the manuscript, I think the title is not appropriated, since none of their data confirm the effect of aqueous alteration. Some for figure 8, which is great but not supported by the results.

[Replies] The comment was accepted and the title was revised to include the specific names of the hydrophilic organic molecule groups (hydroxy acids and carboxylic acids). We have already mentioned that Figure 8 shows the "hypothetical concept summary" supported by the data in this paper and the previous report (Naraoka+, Nakamura+2023, Noguchi+2023). Hence, the figure caption was modified as "Conceptualization of hypothesis: Aqueous alteration in primordial hydrophilic...". In addition, important reference information related to that point was updated.

[Ref.]

Schmitt-Kopplin, P. et al. Soluble organic matter Molecular atlas of Ryugu reveals cold hydrothermalism on C-type asteroid parent body. *Nature Commun.* **14**, Article number: 6525 (2023).

Yoshimura, T. et al. Chemical evolution of primordial salts and organic sulfur molecules in the asteroid (162173) Ryugu. *Nature Commun.* **14**, Article number: 5284 (2023).

Zeichner, S. S. et al. Polycyclic aromatic hydrocarbons in samples of Ryugu formed in the interstellar medium. *Science* **382**, 1411-1416.

Specific comments from line to line or in figures:

Main text:

Figure 1: the figure is very nice, but do contain a lot of information, which are not considered/discussed here. Is it useful ?

[Replies] Reviewers #1 and #2 approve of the overview and usefulness of Figure 1. The authors do not consider this paper from a narrow perspective, but rather make broad observations at various scales to pursue a group of hydrophilic organic molecules that are relatively fragile and difficult to evaluate. It can be understood that the response to the authors' main point is the overall evaluation of reviewer #1.

Lin 87: I don't think the question regarding Ryugu's role in the solar system history is relevant.

[Replies] The phrase was revised to "*What is the role of carbonaceous asteroids in the the Solar System history?*"

Figure 2: pyruvic acid is not mentioned in the legend.

[Replies] Pyruvate was added to the caption.

Line 113: why the aqueous alteration should be only recorded by the polar organic molecules ? please rephrase.

[Replies] Revised as "...aqueous alteration could have been recorded in these hydrophilic organic molecules".

Line 127: write the analytical technic from which you obtained the identification of hydroxyl acids. And figure 2 should be cited here.

[Replies] Revised as "We first identified highly diverse hydroxy acids and hydrophilic molecular groups in the hot water extracts by CE-HRMS (Figure 2)".

Line 128: sample name (#7-1) the name of the sample appears from nowhere, lacking an explanation (we have to go in the SI, looking at Fig S2 !). once again, you need to appeal to a wide audience for a Nature paper.

[Replies] Deleted the sample name (#7-1) in the main text, except in figure and table caption.

Line 128: After giving the name of the hot water fraction, it would be a good idea to refer to figure S3 detailing all the extracts.

[Replies] We added the sample name (A0106 & C0107) and representative compound name for hydroxy acids (e.g., glycolic acid, lactic acid), di-carboxylic acids (e.g., oxalic acid, maleic acid) and other molecules (e.g., mevalonic acid, citric acid) on the paragraph with the cite (Naraoka+2023; see Method).

Line 129: Are the authentic standards those in figure S4? Also in the figure 3-A, please add the standard signal. It is stated in the text that identification is possible using the retention time and the exact mass. However, if we refer to figure S4 and compare the retention times of the standards with figure 3-A, it is impossible to identify the compounds in Ryugu extracts.

Line 129, Also, what is an authentic standard ? can we have factice standards ??

[Replies]

Based on the comment, we have also added reference standard to the chart in Figure 3-A.

We have included an additional description, “we used an internal standard (iSTD) substance to ensure the identification on the migration time as previous report (Sasaki+2019).” As shown in Figure 3, the Murchison meteorite reference and its standardization ensured the molecular identifications in this report. The standard mixture including the working reagents for migration-time alignment (e.g., AM1, AM2, AM3, AM4, and AM5), and an internal standard (iSTD) for anion analysis (ISA) were prepared concurrently from an HMT metabolomics kit (Human Metabolome Technologies Inc., Tsuruoka, Japan) (Sasaki+2019).

Figure 3-C, please specify in legend which molecules were used in the “ α -hydroxy- acids (Cn) in Murchison”.

[Replies] Revised as “...for the concentration of short-chain α -hydroxy acids (i.e., glycolic acid, lactic acid, 2-hydroxybutyric acid, 2-hydroxyvaleric acid)...

Line 133: How was the quantification carried out? External calibration on standards is a very uncertain method due to the presence of salts and the famous matrix effect on the detected compounds.

[Replies] Quantification was performed using reference standards of the identified molecules (Sasaki+2019; Sugimoto+2010). The analytical assurance procedure for the matrix effect is as described above.

Table S1: Please specify error bars, also specify in the legend abbreviations such as "n.d.". Not all the compounds in figure 2 are shown, but some that are never mentioned are.

[Replies] Abbreviations such as "n.d." are also clearly indicated in the legend. We also added the description in the caption as, “We note that the quantitative error with the organic acid mixture solution (average mass number: 120.2 ± 40.6 , $n = 5$) and amino acid mixture solution (average mass number: 132.2 ± 31.0 , $n = 27$) was better than $94.9 \pm 9.4\%$ ($n = 5$) and $90.4 \pm 12.3\%$ ($n = 27$), respectively, as the accuracy of the analytical condition.

“

Line 139: pyruvic acid is not included in table S1

[Replies] Revised the Table S1 with the profile of pyruvic acid.

Line 147-148: Urea and glycoamine are not included in table S1.

[Replies] Revised the Table S1 with the profile of urea and glycoamine.

Line 151: please specify where in the SI. In the paragraph "Spectroscopic FTIR references for the soluble components", these bands are not mentioned, nor are they mentioned in figure S9, although it would appear that they are mentioned in figure S13 (which is in fact never cited).

[Replies] In accordance with the comment, we have clearly indicated as "grain-scale observation; Supplementary Figure S13".

Figure 4-A: Does the x-axis "dicarboxylic acid" take into account all the compounds detailed on lines

[Replies] The names C2 through C8 are shown in the caption as follows, "(A) Dicarboxylic acid profiles (i.e., C₂, Oxalic acid; C₃, Malonic acid; C₄, Succinic acid; C₅, Glutaric acid; C₆, Adipic acid; C₇, Pimelic acid; C₈, Azelaic acid)".

157-158? Please specify in the legend.

[Replies] Revised as "dicarboxylic acids (e.g., C₂, Oxalic acid; C₃, Malonic acid; C₄, Succinic acid; C₅, Glutaric acid; C₆, Adipic acid)".

Figure 4-B: The two-enol forms of malonic acid are in fact the same, since the molecule is symmetrical.

[Replies] That's true. It was mentioned in the caption of Figure 4-B that "The two enol forms of malonic acid (MA) are symmetric and in fact identical."

Line 168- Please refer to figure 4-B.

[Replies] Revised as suggested.

Line 172: please smooth this sentence, as other ways of forming acetic acid can take place in these objects and therefore modify the malonic/acetic ratio, as the formation pathway from malonic acid or acetic acid under aqueous alteration conditions is not the only one.

[Replies] This sentence was revised as follows, “The relative abundance of malonic acid in dicarboxylic acids is an indicator of the aqueous alteration process recorded on the asteroid Ryugu. In fact, the relative abundance of malonic acid is an order of magnitude lower than CM meteorites (e.g., Murchison and Murray as shown in Figure 4-A), suggesting a different history.”

Figure 5: in the legend (line 635), Table 1 does not exist; the reference should be Table S1. Please clearly state what is new and your work here. Are all the plots in this figure really relevant to this study ? IB is not explained.

[Replies] Revised as suggested for Table S1. Firstly, we added the scientific justification as “Please note that the Ryugu sample is of scientific value as a surface (TD1) and subsurface sample (TD2) from the carbonaceous asteroid (Tachibana+2022; Nishiizumi+2022). In this report, we evaluated the hydrophilic organic molecules in both the surface aggregate (A0106 from TD1 site) and subsurface aggregate samples (C0107 from TD2 site) as follows.”

Secondly, as stated by the reviewer #1, the new scientific value is evidence that a group of water-soluble and fragile organic molecules is definitely present on the carbonaceous asteroid. In the process of proving this, the authors had to comprehensively evaluate the initial bulk (IB) properties of the light elements and the two sampling sites. IB (Initial bulk) abbreviation is shown in the caption.

Line 188: Where are the data for the figure 5-F about the urea molecules ?

[Replies] The raw data in Figure 5-F were compiled into Table S1.

Line 190: ammonium ions has been proposed has the main salt component of comet 67P, please refer to this study also.

[Replies] Revised as suggested with the reference.

[Ref.]

Altwegg, K. et al. Evidence of ammonium salts in comet 67P as explanation for the nitrogen depletion in cometary comae. *Nature Astronomy* **4**, 533-540 (2020).

From line 197 to 211, the authors discussed the detection of amino acids in CM and CI, compared to Ryugu, as these class of meteorites where the same. This is not true, and it is quite known now

that parent bodies of CM and CI has not accreted the same pool of material from the nebula, including the organics. And of course, they has not the same aqueous/thermal history. So it is questionable to compare quantities and identification of amino acids from group to group.

[Replies] Of course, the authors understand well that parent bodies of CM and CI has not accreted the same pool of material from the nebula, including the organics. Except for previous reports (Naraoka+2023 and subsequent reports), no one has verified the comparative evaluation of CI and CM with Ryugu presented here for the series of hydrophilic organic molecules. Here, we state that this is the first report in which the other reviewers (#1 & #2) have clearly indicated their approval of novelty.

Line 203: the reference Burton is not correctly cited.

[Replies] Revised as suggested for Burton et al. (2014).

Line 237: missing words

[Replies] Revised by deleting the "~" in that sentence to avoid misunderstanding.

Figure 6-A: Are all the molecules detected and reported in table S1 ?

[Replies] The caption of the figure 6-A stated the following, “Hydroxy acids, dicarboxylic and tricarboxylic acids along with other newly identified hydrophilic molecules for a comparison between Ryugu (this study) and CM type....”

Line 244: Misquotation of figure.

[Replies] Revised as “...icy dry and aqueous wet cycles, Figure 1-B) correlated with the molecular evolution between water and organic matter within the cold hydrothermalism (Schmitt-Kopplin+2023).”

Method section:

Line 288: Why is Figure 4 cited here ?

[Replies] The citation for figure 4 has been removed.

Line 275: In this paragraph, add the origin of the different standards and their purity.

[Replies] Additional information on the standard samples was added. Due to the word limit of the main script, reference articles were also used as follows, “We used the most representative carbonaceous meteorite of Murchison (Naraoka+2023; Oba+2023b; Koga+2024), as a reference standard to confirm our qualitative evaluation of the sample matrix effects. The standard mixture including the working reagents for migration-time alignment (e.g., AM1, AM2, AM3, AM4, and AM5), and an internal standard for anion analysis (ISA) were prepared from an HMT metabolomics kit (Human Metabolome Technologies Inc., Tsuruoka, Japan) (e.g., Sasaki+2019; Kami+2013; Sugimoto+2010). “

[Ref.]

Kami, K. et al. Metabolomic profiling of lung and prostate tumor tissues by capillary electrophoresis time-of-flight mass spectrometry. *Metabolomics* **9**, 444-453 (2013).

Koga, T. et al. Abundant extraterrestrial purine nucleobases in the Murchison meteorite: Implications for a unified mechanism for purine synthesis in carbonaceous chondrite parent bodies. *Geochim. Cosmochim. Acta*, **365**, 253–265 (2024).

Line 304: Misquotation of figure, should be Figure 1-G.

[Replies] Revised as suggested for Figure 1-G.

Line 350: Please specify what is referred to in the supplementary information, as figures S-10 to S-14 are never mentioned.

[Replies] Revised as suggested.

Supporting info

Line 771: in Figure S1 only the sample C0107 is presented.

[Replies] We added the sample A0106 in accordance with review comments.

Line 772 to 773, is there a missing word ?

[Replies] We revised as, “We present a concept based on the dimensions of chemical resolution and chemical variation (Figure S2).”

Figure S3: in the legend, please add the description for C.

[Replies] We revised as “(B, C)...”.

Figure S5: line 950 in the legend, misquotation of figure it should be Figure S9 rather than Figure 9.

[Replies] We revised as “...shown in Fig. S7 and Fig. S8, respectively.”

Line 812: the name “remaining nitrogen” is misleading. The amount of nitrogen was summed after each successive extraction and taken as the total amount in the bulk (initial bulk as 100%). So this is not the remaining nitrogen but the nitrogen extracted by each solvent extraction. The remaining nitrogen is the one that remains in the non-soluble part (IOM).

[Replies] To avoid misleading, we deleted “that remains” in the line.

Line 836: The extract #7-1 is called hot H₂O extract in the text but H₂O-HCl extracts in the legend of Figure S7.

[Replies] That’s true. We appended the extraction fraction numbers (#7-1, #7-2: after Naraoka+2023; Parker+2023; Yoshimura+2023) to the caption of Figure S7.

Figure S7, please add name of the axes for all “lines of plots”.

[Replies] We added the explanation as “The horizontal and vertical axes represents for wavelength (nm) and absorbance (arbitrary unit), respectively.”

Figure S9, name of the axes ?

[Replies] We added the axes on the Figure S9.

Line 967: For the formic acid extract, please add the blank in the Figure S8.

[Replies] We revised in the caption as, “Here, we note that the blank run of formic acid (purity >99%, FUJIFILM Wako pure chemical corporation) was not significant during the sequential analysis (i.e., the instrumental background level of the SALDI system).”

Figure S12-B: The Kovats relation is linear for $\log(\text{tr}')=f(\text{C-number})$, the current representation is wrong. Please correct.

[Replies] We appreciate the comment and removed Fig.S12-B because it is sufficient validation data for the molecular identification. Here, we note that $\log(t_r')=f(\text{C-number})$ with “constant” thermal gradient.

Reviewer #3 (Remarks to the Author):

Dear All,

I appreciate the improvements made to the paper, which has now matured for publication. It would have been ideal to have started with this version as the initial submission.

I have noticed that you have resolved the analytical issues and made changes to some figures, this is great. I have not had the time to verify if all the figures are correctly cited in the text so I hope that they are properly cited.

Line 115-116: the addition of the end of the sentence is helping a lot to understand the goal of this paper.

I suggest to move the sentence line 124 to 125 to this beginning of paragraph, and to link the sentence added about Naraoka's paper, I suggest something as follow :

In this report, we determine the molecular diversity of polar organic molecules extracted from the first contact between hot water and pristine Ryugu samples, and report the unique color characteristics of the sequentially extracted fractions with the systematic variation in ^{13}C - and ^{15}N -isotopic profiles.

If indigeneous....[...]

We use this information to interpret the aqueous alteration processes that asteroid Ryugu has experienced in complement to the study by Naraoka et al. (2023), who reported organic molecular diversity from initial bulk (IB) to insoluble organic matter (IOM) in the sequential extraction process using hydrophilic to hydrophobic solvents.

Line 118. This is not "prebiotic" here. Please remove.

Line 186 " than in other CM meteorites"

I deeply appreciate the new figure 4 and the new explication in the text. May be one point, which is not clear here, is the origin of malonic acid, do you supposed that the molecule is already present in the initial OM of ryugu's parent body ? or has been synthesized in the parent body by aqueous alteration and then has decomposed faster due to tautomerisation? then In this case, what are the precursors of malonic acid and the chemical mechanism compared to other dicarboxylic acid formation ? is this could explain the different abundance of malonic acid between ryugu and murchison ?

Line 265, I think references are need here.

Line 266 "will be further elucidated", is the future time used correct ?

Line 272, still not prebiotic. Prebiotic chemical evolutions are those that existed on Earth before the emergence of biotic systems, since no life exist on asteroids, the term prebiotic cannot be used here. It is already being used excessively in many incorrect contexts.

Replies to Reviewer's comments on the manuscript # NCOMMS-23-43936A

We again appreciate the 2nd reviewing comments on our revised manuscript (# NCOMMS-23-43936A) entitled " Primordial aqueous alteration recorded in water-soluble organic molecules of hydroxy acids and carboxylic acids from the carbonaceous asteroid (162173) Ryugu". We carefully read **the comments in blue** (#3: anonymous) and modified all of the concerns in the revised manuscript. The #3 reviewer's comments are improved **in red** on the main manuscript and supporting information. All comments and the corresponding feedbacks will also be made public on the journal site.

Reviewer #3 (Remarks to the Author):

I appreciate the improvements made to the paper, which has now matured for publication. It would have been ideal to have started with this version as the initial submission. I have noticed that you have resolved the analytical issues and made changes to some figures, this is great. I have not had the time to verify if all the figures are correctly cited in the text so I hope that they are properly cited.

[Replies] The authors are encouraged by the above comments. We rechecked all figures and tables for correspondence in the text. All references were also numbered for main text and supplementary information. In the 2nd revised manuscript, several references were updated to the latest information.

[Refs.]

Hashizume, K. The Earth atmosphere-like bulk nitrogen isotope composition obtained by stepwise combustion analyses of Ryugu return samples. *Meteorit. & Planet. Sci.*, doi: 10.1111/maps.14175. (2024).

Furusho, A. et al. Enantioselective Three-Dimensional High-Performance Liquid Chromatographic Determination of Amino Acids in the Hayabusa2 Returned Samples from the Asteroid Ryugu. *J. Chromatogr. Open.* (2024) (in press).

Line 115-116: the addition of the end of the sentence is helping a lot to understand the goal of this paper. I suggest to move the sentence line 124 to 125 to this beginning of paragraph, and to link the sentence added about Naraoka's paper, I suggest something as follow:

In this report, we determine the molecular diversity of polar organic molecules extracted from the first contact between hot water and pristine Ryugu samples, and report the unique color characteristics of the sequentially extracted fractions with the systematic variation in ¹³C- and

¹⁵N-isotopic profiles. If indigenous....[...]... We use this information to interpret the aqueous alteration processes that asteroid Ryugu has experienced in complement to the study by Naraoka et al. (2023), who reported organic molecular diversity from initial bulk (IB) to insoluble organic matter (IOM) in the sequential extraction process using hydrophilic to hydrophobic solvents.

[Replies] We appreciate the constructive suggestions. The context was refined again according to the comments.

Line 118. This is not “prebiotic” here. Please remove.

[Replies] Removed it and replaced as “abiotic”.

Line 186 “ than in other CM meteorites”

[Replies] Revised as suggested.

I deeply appreciate the new figure 4 and the new explication in the text. May be one point, which is not clear here, is the origin of malonic acid, do you supposed that the molecule is already present in the initial OM of ryugu’s parent body ? or has been synthesized in the parent body by aqueous alteration and then has decomposed faster due to tautomerisation? then In this case, what are the precursors of malonic acid and the chemical mechanism compared to other dicarboxylic acid formation ? is this could explain the different abundance of malonic acid between ryugu and munchison ?

[Replies] The question of what is the precursor of organic acids has been an issue that has not been resolved in previous reports (e.g., Peltzer+1984; Shimoyama & Shigematsu, 1994) and current organic description from Ryugu samples (Naraoka+2023; Schmitt-Kopplin+2023). To the best of our knowledge, it is difficult to derive a definitive answer simply by assuming photochemical reactions, radical reactions, or hydrothermal ionic reactions (e.g., Ehrenfreund+2001; Öberg, 2016; Glavin+2018). Likewise the quest of reviewer#3, we anticipate clarification through the additional analysis of Ryugu and asteroid Bennu samples (e.g., Galvin+2024). Then, it should be noted that the organic analysis of the asteroid Bennu is a valuable opportunity to consider the science consequences of this study.

[Refs.]

Öberg, K. I. Photochemistry and astrochemistry: Photochemical pathways to interstellar complex organic molecules. *Chem. Rev.* **116**, 9631–9663 (2016).

Glavin, D. P. et al. Investigating the impact of x-ray computed tomography imaging on soluble organic matter in the Murchison meteorite: Implications for Bennu sample analyses. *Meteorit. & Planet. Sci.* **59**, 105–133 (2024).

Line 265, I think references are need here.

[Replies] We agree this point and added the references as correspondingly.

[Refs.]

Yamaguchi, A. et al. Insight into multi-step geological evolution of C-type asteroids from Ryugu particles. *Nature Astron.* **7**, 398–405 (2023).

Fujiya, W. et al. Carbonate record of temporal change in oxygen fugacity and gaseous species in asteroid Ryugu. *Nature Geosci.* **16**, 675–682 (2023).

Line 266 “will be further elucidated”, is the future time used correct ?

[Replies] Deleted “further” as suggested.

Line 272, still not prebiotic. Prebiotic chemical evolutions are those that existed on Earth before the emergence of biotic systems, since no life exist on asteroids, the term prebiotic cannot be used here.

[Replies] Deleted the term of “prebiotic”.